# Identification of a phosphorylation site on Ulk1 required for genotoxic stress-induced alternative autophagy

Satoru Torii [1✉], Hirofumi Yamaguchi[1], Akira Nakanishi[2], Satoko Arakawa[1], Shinya Honda[1], Kenta Moriwaki[3], Hiroyasu Nakano [4] & Shigeomi Shimizu [1✉]

Alternative autophagy is an autophagy-related protein 5 (Atg5)-independent type of macroautophagy. Unc51-like kinase 1 (Ulk1) is an essential initiator not only for Atg5-dependent canonical autophagy but also for alternative autophagy. However, the mechanism as to how Ulk1 differentially regulates both types of autophagy has remained unclear. In this study, we identify a phosphorylation site of Ulk1 at Ser[746], which is phosphorylated during genotoxic stress-induced alternative autophagy. Phospho-Ulk1[746] localizes exclusively on the Golgi and is required for alternative autophagy, but not canonical autophagy. We also identify receptor-interacting protein kinase 3 (RIPK3) as the kinase responsible for genotoxic stress-induced Ulk1[746] phosphorylation, because RIPK3 interacts with and phosphorylates Ulk1 at Ser[746], and loss of RIPK3 abolishes Ulk1[746] phosphorylation. These findings indicate that RIPK3-dependent Ulk1[746] phosphorylation on the Golgi plays a pivotal role in genotoxic stress-induced alternative autophagy.

[1] Department of Pathological Cell Biology, Medical Research Institute, Tokyo Medical and Dental University (TMDU), 1-5-45 Yushima, Bunkyo-ku, Tokyo 113-8510, Japan. [2] Department of Molecular Genetics, Medical Research Institute, Tokyo Medical and Dental University (TMDU), 1-5-45 Yushima, Bunkyo-ku, Tokyo 113-8510, Japan. [3] Department of Cell Biology, Graduate School of Medicine, Osaka University, 2-2 Yamadaoka, Suita, Osaka 565-0871, Japan. [4] Department of Biochemistry, Toho University School of Medicine, 5-21-16 Omori-Nishi, Ota-ku, Tokyo 143-8540, Japan. ✉email: toripcb@tmd.ac.jp; shimizu.pcb@mri.tmd.ac.jp

Macroautophagy (hereafter referred to as autophagy) is a catabolic process in which cellular contents are degraded[1,2]. Autophagy plays a pivotal role in a wide variety of physiological and pathological situations. The molecular basis of starvation-induced autophagy has been extensively studied, in which autophagic membranes originate from the endoplasmic reticulum (ER) or mitochondria-associated ER membrane and functional complexes containing autophagy-related (Atg) proteins drive the formation of autophagosomes[2,3]. Atg proteins are categorized into the following five functional groups: Unc-51 like autophagy activating kinase (Ulk1) and its regulators, the class III lipid kinases producing phosphatidylinositol 3-phosphate (PI3K), Atg9 complexes, the Atg5 conjugation system, and the microtubule-associated protein light chain 3 (LC3) conjugation system. After their generation, autophagosomes fuse with lysosomes to form autolysosomes by a mechanism dependent on syntaxin 17 (Stx17)[4]. In addition to this canonical autophagy, several types of noncanonical autophagy have been reported[5–7]. Atg5-independent alternative macroautophagy (hereafter described as alternative autophagy), which we discovered previously[5], is one such type of autophagy. This autophagy machinery is also named Golgi membrane-associated degradation (GOMED)[8]. Unlike canonical autophagy, autophagosomal membranes are derived from the *trans*-Golgi membrane[5,8] in alternative autophagy, and the mechanism involves Ulk1 and PI3K complexes, but not Atg9 complexes, Stx17[9], nor the Atg5 or LC3 conjugation system. Canonical and alterative autophagy are used differently in a stimulus/context-dependent manner, i.e., starvation induces mostly canonical autophagy, whereas genotoxic stress induces both canonical and alternative autophagy[5]. Furthermore, in the terminal differentiation of erythrocytes, alternative and canonical autophagy eliminate different substrates, i.e., mitochondria and ribosomes, respectively[10].

Despite morphological similarities, molecules involved in canonical and alternative autophagy are mostly different. However, Ulk1 is a serine/threonine kinase and a homolog of yeast Atg1 that is involved in both pathways at the initial step[5,10]. Ulk1 contains various phosphorylation sites, and their phosphorylation status regulates canonical autophagy[3]. Under unstimulated conditions, Ulk1 is phosphorylated at $Ser^{637}$ (amino acid residues are described according to mouse Ulk1, unless otherwise described) and $Ser^{757}$ by mammalian target of rapamycin complex 1[11,12], by which Ulk1 is inactivated. Upon starvation and genotoxic stress, Ulk1 is dephosphorylated at $Ser^{637}$ by protein phosphatase 2A[13] and protein phosphatase, $Mg^{2+}/Mn^{2+}$-dependent 1D (PPM1D)[14], respectively. This dephosphorylation facilitates Ulk1 translocation to preautophagosomal membranes and activates its kinase activity, both of which are required for the induction of canonical autophagy. AMP-activated protein kinase (AMPK) was also reported to phosphorylate Ulk1 at $Ser^{317}$, $Ser^{467}$, $Ser^{555}$, $Ser^{777}$, and $Thr^{574}$ upon starvation[3,12,15,16]. Insulin-Akt signaling also induces Ulk1 phosphorylation at $Ser^{774}$ and blocks rapamycin-induced autophagy[16]. Thus, Ulk1 activity is regulated by the status of its multiple phosphorylation sites. Because more than 70 phosphorylation sites have been reported in Ulk1, additional kinases are expected to be involved in the regulation of Ulk1 activity and the resulting cellular events.

In the mechanism of alternative autophagy, Ulk1 functions at the initial step[5,10], because knockdown of Ulk1 suppressed the generation of isolation membranes. Unlike canonical autophagy, however, the mechanism of Ulk1 activation remains unclear. Therefore, to address this issue, we search for Ulk1 phosphorylation sites required for the induction of alternative autophagy. In this study, we identiy a phosphorylation site of Ulk1 at $Ser^{746}$, which is required for the initial step of alternative autophagy, but not canonical autophagy, upon stimulation of genotoxic stress.

We further identify receptor-interacting serine-threonine kinase 3 (RIPK3) as the kinase responsible for Ulk1 phosphorylation at $Ser^{746}$. We also find that Ulk1 phosphorylated on $Ser^{746}$ is localized on the Golgi, where isolation membranes are generated. Taken together, these data indicate that RIPK3-dependent Ulk1 phosphorylation at $Ser^{746}$ is required for alternative autophagy, but not canonical autophagy.

## Results

**DNA damage induces Ulk1 phosphorylation at $Ser^{746}$.** Ulk1 is a crucial molecule not only for canonical autophagy but also for alternative autophagy[5]. Unlike canonical autophagy, how Ulk1 is activated during alternative autophagy remains unknown. Because the properties and functions of Ulk1 are largely dependent on the phosphorylation status of each amino acid residue, we first investigated the sites of Ulk1 phosphorylation during alternative autophagy induced by etoposide, a DNA-damaging reagent and strong inducer of alternative autophagy. To this end, Ulk1 was immunoprecipitated with an anti-Ulk1 antibody from untreated and etoposide-treated Atg5 knockout (hereafter referred to as $Atg5^{KO}$) mouse embryonic fibroblasts (MEFs), digested with trypsin, and then analyzed by LC–MS/MS. We detected various reported phosphorylation sites of Ulk1, such as $Ser^{317}$, which is an AMPK target that is known to be phosphorylated during alternative autophagy[17]. In addition, we identified phosphorylation sites of Ulk1 at $Thr^{10}$, $Ser^{297}$, $Ser^{300}$, $Ser^{302}$, and $Ser^{783}$ in untreated $Atg5^{KO}$ MEFs, and at $Ser^{308}$, $Ser^{314}$, $Ser^{494}$, and $Ser^{746}$ in etoposide-treated $Atg5^{KO}$ MEFs.

Among these phosphorylation sites, we focused on $Ser^{746}$ (Fig. 1a), because when various phosphodeficient Ulk1 mutants were expressed at equivalent levels in Atg5/Ulk1 double-knockout ($Atg5/Ulk1^{DKO}$) MEFs (Supplementary Fig. 1a, b), most Ulk1 mutants, but not mutant Ulk1 (S746A), recovered the ability to perform alternative autophagy (Supplementary Fig. 1c, d). The $Ser^{746}$ residue (corresponding to $Ser^{747}$ of human Ulk1) is conserved in Ulk1, but not Ulk2, of higher vertebrates (Supplementary Fig. 1e). To analyze the role of Ulk1 phosphorylated at $Ser^{746}$ (p-Ulk1$^{746}$), we generated a specific antibody against p-Ulk1$^{746}$ that could be used for immunostaining and immunoprecipitation, but not for western blotting. To confirm the phosphorylation of $Ser^{746}$ of Ulk1 upon etoposide treatment, we immunoprecipitated p-Ulk1$^{746}$ using its specific antibody and analyzed the amount of precipitation by western blotting using an anti-Ulk1 antibody. As indicated, the level of endogenous p-Ulk1$^{746}$ was increased in $Atg5^{KO}$ MEFs, but not in $Atg5/Ulk1^{DKO}$ MEFs, upon etoposide treatment (Fig. 1b). The p-Ulk1$^{746}$ signal was completely abolished by the addition of recombinant phosphatase during the immunoprecipitation (Supplementary Fig. 2), indicating that the immunoprecipitation occurred in a phosphorylation-dependent manner. When we expressed HA-Ulk1 (wild-type; WT) in $Atg5/Ulk1^{DKO}$ MEFs, exogenous p-Ulk1$^{746}$ signals were also increased, whereas it was not observed upon the expression of the S746A phosphodeficient mutant (Fig. 1c), despite mutant Ulk1 being expressed at a higher level than HA-Ulk1 (WT) (Fig. 1c). These data validate the quality of the p-Ulk1$^{746}$-specific antibody and confirmed the etoposide-induced phosphorylation of Ulk1 at $Ser^{746}$. Note that a mobility shift in Ulk1 was observed in etoposide-treated cells on SDS–PAGE (Fig. 1b, c), which might be due to the dephosphorylation of Ulk1 at other residues, such as $Ser^{637}$, as previously described[14]. Analysis of $Ser^{637}$ dephosphorylation is described later.

We also investigated the subcellular localization of p-Ulk1$^{746}$ by immunostaining using this antibody. Consistent with immunoprecipitation-western blotting, a strong p-Ulk1$^{746}$ signal

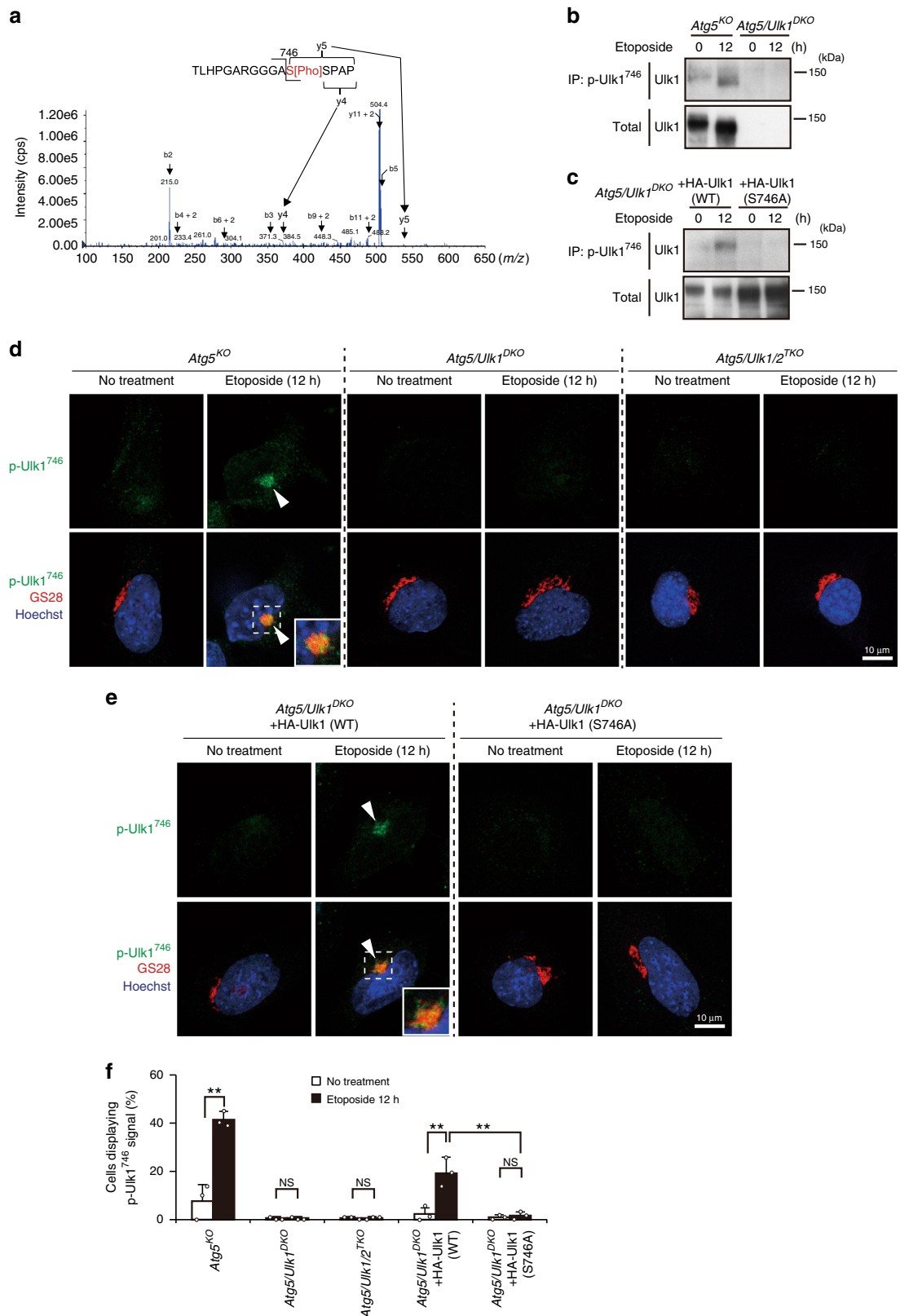

appeared in $Atg5^{KO}$ MEFs upon etoposide treatment (Fig. 1d, f) in a time-dependent and dose-dependent manner (Supplementary Fig. 3). However, these signals were not observed in $Atg5/Ulk1^{DKO}$ MEFs and Atg5/Ulk1/Ulk2 triple-knockout ($Atg5/Ulk1/2^{TKO}$) MEFs (Supplementary Fig. 4) upon etoposide treatment (Fig. 1d, f). Furthermore, HA-Ulk1 (WT)-expressing,

but not HA-Ulk1 (S746A) mutant-expressing $Atg5/Ulk1^{DKO}$ MEFs showed p-Ulk1$^{746}$ signals after etoposide treatment (Fig. 1e, f). These findings validate the usefulness of our antibody for immunofluorescence experiments, and again confirmed the etoposide-induced phosphorylation of Ulk1 at Ser$^{746}$. Interestingly, p-Ulk1$^{746}$ signals merged almost completely with

**Fig. 1 Phosphorylation of Ulk1 at Ser$^{746}$ and its Golgi localization upon etoposide treatment. a** Identification of an Ulk1 phosphorylation site. Ulk1 was immunoprecipitated with the anti-Ulk1 antibody from etoposide-treated Atg5$^{KO}$ MEFs and subjected to trypsin digestion. The tryptic digests were analyzed by LC–MS/MS. This mass spectrum yielded a fragment ion spectrum displaying three C-terminal fragment ions (y-type) and seven N-terminal fragment ions (b-type). The result that y5-y4 is about 167 Da, which is equivalent to a phosphoserine, and database searching identified this peptide as TLHPGARGGGAS[Pho]SPAP, the partial sequence (amino acids 735–750) of the Ulk1 protein. **b, c** Phosphorylation of Ulk1 at Ser$^{746}$ by etoposide treatment. The indicated MEFs were treated with 10 μM of etoposide for the indicated times, lysed, and immunoprecipitated with an anti-p-Ulk1$^{746}$ antibody. Immune complexes and total lysates (2.8% input) were analyzed by western blotting using an anti-Ulk1 antibody. **d, e** Induction of the Golgi localization of p-Ulk1$^{746}$ by etoposide treatment. The indicated MEFs were treated with or without 10 μM of etoposide for 12 h, and immunostained with anti-p-Ulk1$^{746}$ and anti-GS28 antibodies. Nuclei were counterstained with Hoechst 33342 (50 ng mL$^{-1}$). Representative images of p-Ulk1$^{746}$ (green; upper panels) and merged images (lower panels) of p-Ulk1$^{746}$ (green), GS28 (red), and Hoechst 33342 (blue) are shown. Magnified images of the areas within the dashed squares are shown in the inset. Arrowheads indicate p-Ulk1$^{746}$ signals. **f** Quantification of cells displaying p-Ulk1$^{746}$ signals. The indicated MEFs were treated with 10 μM of etoposide for the indicated times, and immunostained with an anti-p-Ulk1$^{746}$ antibody. The population of cells with p-Ulk1$^{746}$ signals was calculated ($n \geq 100$ cells in each experiment). Data are shown as the mean ± SD ($n = 3$). Atg5/Ulk1$^{DKO}$ + HA-Ulk1 (WT) no treatment vs. etoposide: $p = 0.0002$. Other exact $p$ values cannot be described since the value is too large ($p > 0.9999$) or small ($p < 0.0001$). Comparisons were performed using one-way ANOVA followed by the Tukey post-hoc test. $*p < 0.05$, $**p < 0.01$; NS: not significant. Source data are provided as a Source Data file.

---

immunofluorescence signals of the Golgi marker GS28 (Fig. 1d, e). The Golgi localization of p-Ulk1$^{746}$ is reasonable because Golgi membranes are the source of alternative autophagy[5].

**Role of Ulk1 Ser$^{746}$ phosphorylation in alternative autophagy.** As we found that etoposide treatment of cells leads to the formation of p-Ulk1$^{746}$ on the Golgi and induces alternative autophagy in an Ulk1-dependent manner, we next analyzed the causal relationship between Ulk1 Ser$^{746}$ phosphorylation and alternative autophagy. To this end, we analyzed alternative autophagy using red-fluorescent protein (RFP)–green-fluorescent protein (GFP) tandem proteins[18]. Autolysosomes are detected as red puncta because GFP fluorescence, but not RFP fluorescence, becomes weak within acidic lysosomal compartments. Correlative light and electron microscopic (CLEM) analysis confirmed the red puncta as autolysosomes (Fig. 2a, Supplementary Fig. 5). As shown in Fig. 2b, red puncta were generated in Atg5$^{KO}$ MEFs upon etoposide treatment. Furthermore, these red puncta were encircled by immunofluorescence signals of Lamp2 (Fig. 2b). These findings were confirmed by the fluorescence intensity line profile (Fig. 2b), indicating the generation of autolysosomes in etoposide-treated Atg5$^{KO}$ MEFs. Unlike Atg5$^{KO}$ MEFs, red puncta were not generated in Atg5/Ulk1$^{DKO}$ MEFs and Atg5/Ulk1/2$^{TKO}$ MEFs after etoposide treatment (Fig. 2c), and were restored by the expression of HA-Ulk1 (WT) (Fig. 2d), but not the HA-Ulk1 (S746A) mutant (Fig. 2e). These results were confirmed by quantitative analysis (Fig. 2f) and were consistent with the induction of p-Ulk1$^{746}$ (Fig. 1f), indicating the requirement of Ulk1 phosphorylation at Ser$^{746}$ for alternative autophagy. Autolysosomes can also be assessed by immunostaining of the lysosomal protein Lamp2, because lysosomal fluorescence increases upon the fusion of lysosomes with autophagic vacuoles. The correspondence of large Lamp2 puncta to autolysosomes was previously shown by CLEM analysis[5,8]. Lamp2 immunostaining showed consistent results with the RFP–GFP tandem protein assay (Fig. 2g, Supplementary Fig. 6). Note that although etoposide activates both autophagy and apoptosis, we performed all the experiments before apoptosis occurred. Furthermore, similar results were obtained regarding autophagy in cells treated with the pan-caspase inhibitor Q-VD-OPh (Supplementary Fig. 7a, b), so that the effect of apoptosis could be disregarded. Taken together, Ulk1 phosphorylation at Ser$^{746}$ is crucial for etoposide-induced alternative autophagy.

**RIPK3 is essential for the Ulk1 Ser$^{746}$ phosphorylation.** Next, we investigated the kinase responsible for this Ulk1

phosphorylation at Ser$^{746}$. Because the target sequence of Ulk1 at Ser$^{746}$ is similar to the sequences of RIPK3 substrates[19] (see section "Discussion"), we focused on RIPK3[20]. Immunoprecipitation analysis showed the interaction between endogenous RIPK3 and endogenous Ulk1 in untreated conditions (Fig. 3a, lane 7), and an increase in RIPK3 binding from 10 h after etoposide treatment (Fig. 3a, lanes 8 and 9). Such an interaction was not observed in Atg5/RIPK3 double-knockout (Atg5/RIPK3$^{DKO}$) MEFs (Supplementary Fig. 8a, Fig. 3a, lanes 10–12). Consistently, incubation of lysates from untreated Atg5$^{KO}$ MEFs, in which Ulk1$^{746}$ is not phosphorylated (Fig. 3b, lane 1), with recombinant RIPK3 in phosphorylation buffer resulted in Ulk1$^{746}$ phosphorylation (Fig. 3b, lane 2). Furthermore, the phosphorylation of Ulk1$^{746}$ was suppressed by the addition of GSK'872, a RIPK3 inhibitor[21] (Fig. 3b, lane 3), indicating that RIPK3 has the potential to phosphorylate Ulk1 at Ser$^{746}$. As expected, etoposide-induced p-Ulk1$^{746}$ signals were observed in Atg5$^{KO}$ MEFs, but not in Atg5/RIPK3$^{DKO}$ MEFs (Fig. 3c, d, Supplementary Fig. 8b), which were restored by the exogenous expression of wild-type RIPK3, but not kinase-deficient RIPK3 (Fig. 3c, d). Consistently, alternative autophagy was observed in Atg5$^{KO}$ MEFs and wild-type RIPK3-expressing Atg5/RIPK3$^{DKO}$ MEFs, but not in Atg5/RIPK3$^{DKO}$ MEFs and kinase-deficient RIPK3-expressing Atg5/RIPK3$^{DKO}$ MEFs (Fig. 3e, f, Supplementary Fig. 8c, d). Pharmacological inhibition of RIPK3 by GSK'872 also suppressed etoposide-induced p-Ulk1$^{746}$ signals (Fig. 3g, h) and alternative autophagy (Supplementary Fig. 8e, f). Consistent with etoposide treatment, camptothecin treatment and ultraviolet C (UVC) exposure, other DNA-damaging treatments also induced the phosphorylation of Ulk1 at Ser$^{746}$ in Atg5$^{KO}$ MEFs, but not in Atg5/RIPK3$^{DKO}$ MEFs (Supplementary Fig. 9). These data indicated that RIPK3 is essential for the genotoxic stress-induced phosphorylation of Ulk1 at Ser$^{746}$ and the subsequent induction of alternative autophagy.

Etoposide induces both canonical and alternative autophagy. Given that p-Ulk1$^{746}$ is crucial for etoposide-induced alternative autophagy, we next investigated whether this phosphorylation is also involved in etoposide-induced canonical autophagy. When MEFs were treated with etoposide, the formation of LC3 puncta was observed in WT MEFs, but not Ulk1/2$^{DKO}$ MEFs (Fig. 4a), indicating the loss of canonical autophagy in Ulk1/2$^{DKO}$ MEFs. The number of cells with LC3 puncta was recovered to a similar level by the expression of the HA-Ulk1 (WT) and the HA-Ulk1 (S746A) mutant (Fig. 4a, b). Because expression of the HA-Ulk1 (S746A) mutant led to the recovery of canonical autophagy but not alternative autophagy, Ulk1$^{746}$ phosphorylation is not involved in etoposide-induced canonical autophagy. Consistent

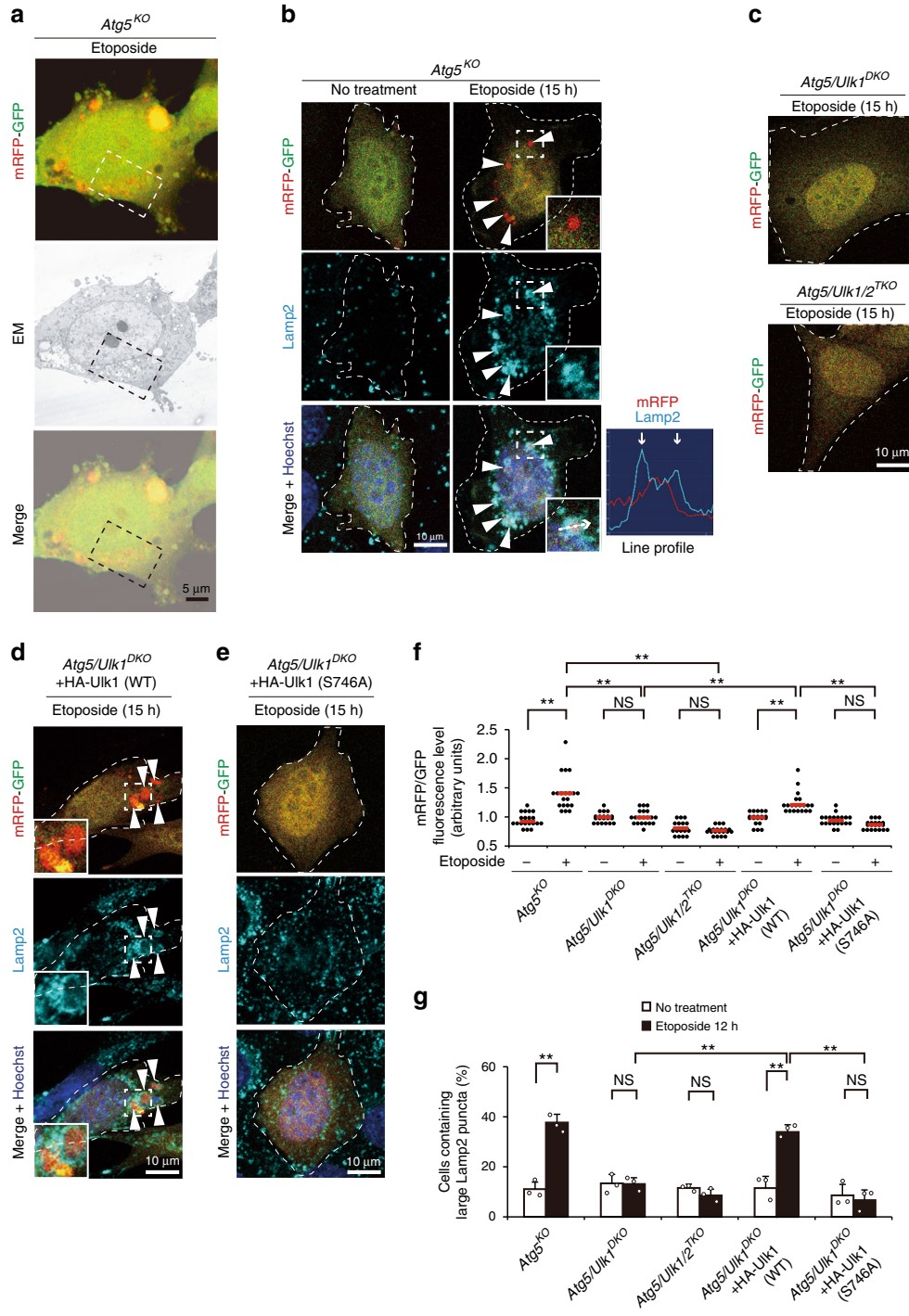

results were obtained when canonical autophagy was assessed by the formation of red puncta in mCherry-GFP-LC3-expressing cells (Supplementary Fig. 10a, b). Autophagy flux analysis of LC3-II and p62 using bafilomycin A1 confirmed the equivalent induction of canonical autophagy (Fig. 4c). Therefore, Ulk1[746] phosphorylation is required for etoposide-induced alternative autophagy (Fig. 2), but not canonical autophagy. Consistently, despite RIPK3 being essential for etoposide-induced alternative autophagy (Fig. 3), canonical autophagy was induced to a similar level in WT MEFs and *RIPK3*[KO] MEFs upon etoposide treatment (Fig. 4d–f, Supplementary Fig. 10c, d), indicating that RIPK3-dependent Ulk1 phosphorylation at Ser[746] is not involved in etoposide-induced canonical autophagy. Absence

of the involvement of Ulk1[746] phosphorylation in canonical autophagy was confirmed because starvation or the addition of rapamycin, both of which strongly induce canonical autophagy but not alternative autophagy, did not generate p-Ulk1[746] signals in WT MEFs (Supplementary Fig. 11a, b), nor *Atg5*[KO] MEFs (Supplementary Fig. 11c, d). Furthermore, starvation-induced canonical autophagy was activated to a similar extent in HA-Ulk1 (WT)-expressing and HA-Ulk1 (S746A)-expressing *Ulk1/2*[DKO] MEFs (Supplementary Fig. 12a, b), as well as in WT MEFs and *RIPK3*[KO] MEFs (Supplementary Fig. 12c, d). These data indicated the crucial role of RIPK3-dependent generation of p-Ulk1[746] in alternative autophagy, but not in canonical autophagy.

**Fig. 2 Requirement of Ulk1 phosphorylation at Ser$^{746}$ in etoposide-induced alternative autophagy. a** CLEM analysis identified red puncta in the autolysosomes of mRFP–GFP-expressing *Atg5$^{KO}$* MEFs. Cells were treated with etoposide (10 μM) for 12 h and observed using fluorescence and electron microscopy. Red puncta were merged with the autophagic vacuoles. Magnified images of the dashed squares are shown in Supplementary Fig. 5. **b–e** The indicated MEFs were transiently transfected with the mRFP–GFP plasmid, and after 24 h, cells were treated with etoposide (10 μM) for 15 h, and immunostained with an anti-Lamp2 antibody. Nuclei were counterstained with Hoechst 33342. Representative images are shown. The dashed lines indicate individual cells. Magnified images of the areas within the dashed squares are shown in the insets. Arrowheads indicate mRFP–GFP red puncta (acidic compartments). In **b**, the fluorescence intensity profile across the white dashed arrows in the inset is shown in the right panel. Note that red puncta were surrounded by Lamp2 signals (arrowheads). **f** Quantification of cells with mRFP–GFP red puncta. The indicated MEFs were treated with or without 10 μM of etoposide for 15 h. The extent of red puncta was analyzed by the RFP intensity/GFP intensity per cell. Each dot indicates the RFP/GFP ratio in a single cell (*n* = 20 cells). Red lines indicate the mean value. *Atg5/Ulk1/2$^{TKO}$* no treatment vs. etoposide: *p* = 0.9986, *Atg5/Ulk1$^{DKO}$* + HA-Ulk1 (S746A) no treatment vs. etoposide: *p* = 0.7931. **g** Quantification of cells with large Lamp2 puncta. The indicated MEFs were treated with or without etoposide (10 μM) for 12 h and immunostained with an anti-Lamp2 antibody. The population of cells with large Lamp2 puncta (≥2 μm) was calculated. (*n* ≥ 100 cells in each experiment). Representative images are shown in Supplementary Fig. 6. Data are shown as the mean ± SD (*n* = 3). *Atg5/Ulk1/2$^{TKO}$*, *Atg5/Ulk1$^{DKO}$* + HA-Ulk1 (S746A), no treatment vs. etoposide: *p* = 0.9797, *p* = 0.9993, respectively. Comparisons were performed using one-way ANOVA followed by the Tukey post-hoc test. \*\**p* < 0.01; NS: not significant. Source data are provided as a Source Data file.

Because RIPK3-dependent p-Ulk1$^{746}$ signals were found to be associated with alternative autophagy (Fig. 2) but not canonical autophagy (Fig. 4), and because we observed these signals in WT MEFs upon etoposide treatment (Fig. 4g, h), we assumed that alternative autophagy would occur in WT MEFs. Consistent results were obtained when WT thymocytes and *RIPK3$^{KO}$* thymocytes were treated with etoposide (Supplementary Fig. 13a, b), confirming the crucial role of RIPK3 in the phosphorylation of Ulk1 at Ser$^{746}$ upon etoposide treatment.

**No crosstalk between necroptosis and alternative autophagy**. In this study, we showed that RIPK3 phosphorylates Ulk1 and thereby induces alternative autophagy upon etoposide treatment. RIPK3 is also known to phosphorylate mixed lineage kinase domain-like (MLKL), and thereby cause membrane rupture, leading to necroptosis when cells are treated with tumor necrosis factor-α (TNF-α), cycloheximide, and the pan-caspase inhibitor zVAD (namely, TCZ[22]) (Fig. 5a). Because RIPK3 is used for both alternative autophagy and necroptosis, we tested whether TCZ stimulation induces alternative autophagy, and whether etoposide induces not only apoptosis but also RIPK3/MLKL-mediated necroptosis. The former possibility was denied because TCZ stimulation did not result in p-Ulk1$^{746}$ phosphorylation (Fig. 5b, Supplementary Fig. 14a) and did not induce alternative autophagy, as assessed by Lamp2 puncta formation (Fig. 5c, Supplementary Fig. 14b). To clarify the latter possibility, we analyzed MLKL phosphorylation and necroptosis induction in etoposide-treated *Atg5$^{KO}$* MEFs. As expected, MLKL activation was not observed (Fig. 5d, lanes 2, 3). Necroptosis was also not induced by etoposide, because cell death was blocked by Q-VD-OPh, but not by Nec-1 and MLKL deletion (Fig. 5e). Therefore, although RIPK3 was activated by etoposide and TCZ treatment, these stimuli induced alternative autophagy and necroptosis, respectively, without any crosstalk (Fig. 5a).

We further analyzed the mechanism of RIPK3 activation after etoposide treatment. Upon TCZ treatment, RIPK1 recruits RIPK3 using their RIP homotypic interaction motif (RHIM) domains (Fig. 5a), which eventually induces RIPK3 homo-oligomerization and generates phospho-RIPK3$^{231,232}$, leading to MLKL activation[23] (Fig. 5a). In marked contrast, RIPK1 was not required for etoposide-induced p-Ulk1$^{746}$ generation, because *RIPK1$^{KO}$* cells normally show p-Ulk1$^{746}$ signals (Fig. 6a, b), unlike *RIPK3$^{KO}$* MEFs (Fig. 3). *MLKL$^{KO}$* cells also showed normal p-Ulk1$^{746}$ signals (Supplementary Fig. 15). RIPK3 oligomerization with a subsequent reduction in RIPK3 monomers was induced by TCZ (Fig. 6c, lane 3), but not by etoposide treatment (Fig. 6c, lane 2). Furthermore, although a RHIM domain mutant ($^{448}$VQIG$^{451}$–$^{448}$AAAA$^{451}$) did not induce

TCZ-induced necroptosis[23], it still induced equivalent levels of p-Ulk1$^{746}$ signals (Fig. 6d, e) and alternative autophagy (Fig. 6f, g) in *Atg5/RIPK3$^{DKO}$* MEFs. Deletion mutant analysis supported that kinase domain, but not RHIM domain, was required for RIPK3–Ulk1 interaction (Supplementary Fig. 16). Moreover, unlike TCZ treatment, etoposide did not generate phospho-RIPK3$^{231,232}$ signals (Fig. 6h, i, lanes 1–3). Taken together, etoposide treatment activates RIPK3 via a completely different mechanism from TCZ treatment.

**Requirement of Ulk1 Ser$^{637}$ dephosphorylation**. How does etoposide induce RIPK3-mediated Ulk1$^{746}$ phosphorylation? We previously reported that etoposide induced the dephosphorylation of Ulk1 at Ser$^{637}$, a different phosphorylation site, in a manner dependent on p53 and PPM1D[14]. We thus investigated whether Ulk1$^{637}$ dephosphorylation is required for Ulk1$^{746}$ phosphorylation and alternative autophagy. To this end, we expressed a phospho-mimetic (S637D) and a phosphodeficient (S637A) Ulk1 mutants in *Atg5/Ulk1$^{DKO}$* MEFs (Fig. 7a). The expression of WT Ulk1 and the S637A mutant, but not the S637D mutant, restored the phosphorylation of Ulk1 at Ser$^{746}$ (Fig. 7b, c), as well as alternative autophagy (Fig. 7d, Supplementary Fig. 17), despite being expressed at levels similar to those of HA-Ulk1 (Fig. 7a). This etoposide-induced dephosphorylation at Ser$^{637}$ of Ulk1 occurs via p53 and downstream of PPM1D (ref. [14] and Fig. 7e). Consistent with the results of experiments using Ulk1 mutants of Ser$^{637}$, etoposide-induced phosphorylation of Ulk1 at Ser$^{746}$ (Fig. 7f, g) and alternative autophagy (Fig. 7h, i, Supplementary Fig. 18) were observed in *Atg5$^{KO}$* MEFs, but not in *Atg5/p53$^{DKO}$* MEFs and *Atg5/PPM1D$^{DKO}$* MEFs. These data indicate that p53/PPM1D-dependent dephosphorylation of Ulk1 at Ser$^{637}$ is required for the RIPK3-induced phosphorylation of Ulk1 at Ser$^{746}$ and the subsequent induction of alternative autophagy. This notion is supported by the fact that expression of the HA-Ulk1 (S637A) mutant resulted in Ulk1$^{746}$ phosphorylation and the induction of alternative autophagy in *Atg5/PPM1D$^{DKO}$* MEFs (Supplementary Fig. 19). This was further confirmed by the induction of alternative autophagy upon the expression of the HA-Ulk1 (S637A and S746D) mutants (PPM1D-dephosphomimetic and RIPK3-phosphomimetic, respectively) without any stimuli (Supplementary Fig. 20). Collectively, the p53–PPM1D axis is required for p53–RIPK3-dependent Ulk1$^{746}$ phosphorylation and alternative autophagy.

We further elucidated how RIPK3 is activated upon etoposide treatment. Because western blot analysis showed the upregulation of RIPK3 in *Atg5$^{KO}$* MEFs in response to etoposide treatment (Fig. 6i, lanes 2 and 3), we suspected the involvement of p53, a master regulator of genotoxic stress. As expected, RIPK3 upregulation was observed in *Atg5$^{KO}$* cells (Fig. 8a). However,

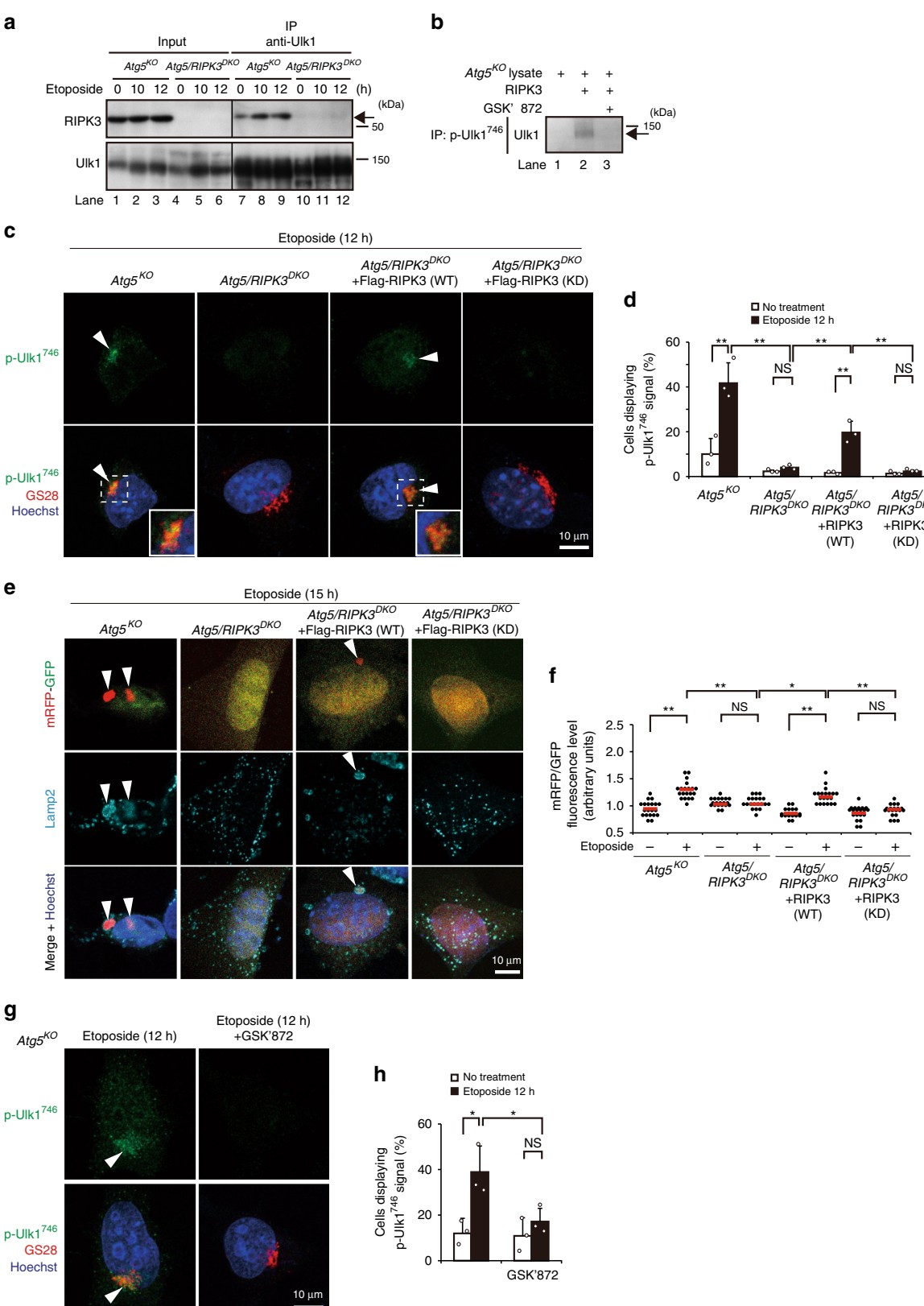

RIPK3 expression was dramatically suppressed in Atg5/p53 double-knockout ($Atg5/p53^{DKO}$) MEFs (Fig. 8a). Very weak expression of RIPK3 was confirmed using different $p53^{KO}$ MEFs (Supplementary Fig. 21a) and by the analysis of mRNA levels (Fig. 8b). Thus, RIPK3 activation upon genotoxic stress was

thought to be induced by p53-dependent transcriptional upregulation. The importance of this RIPK3 upregulation was confirmed by the results that the simple expression of RIPK3, but not that of kinase-deficient RIPK3, was sufficient in generating p-Ulk1[746] in HA-Ulk1 (S637A)-expressing $Atg5/Ulk1^{DKO}$ MEFs

**Fig. 3 Involvement of RIPK3 in the phosphorylation of Ulk1 at Ser[746] and alternative autophagy. a** Physical interaction between endogenous Ulk1 and endogenous RIPK3. The indicated MEFs were treated with etoposide (10 μM). Immunoprecipitation was performed with an anti-Ulk1 antibody. Immune complexes and total lysates (2.8% input) were analyzed by western blotting. **b** In vitro kinase assay of RIPK3 for Ulk1 phosphorylation. Total lysates from untreated $Atg5^{KO}$ MEFs were incubated with GST-RIPK3 (1 μg) with or without 2 μM of GSK'872 for 1 h. Then, the extent of p-Ulk1[746] was analyzed by immunoprecipitation–western blotting. The arrow indicates the p-Ulk1[746] signal. **c–h** Requirement of RIPK3 for etoposide-induced phosphorylation of Ulk1 Ser[746] and alternative autophagy. **c, d** Similar experiments to Fig. 1d were performed using the indicated MEFs. In **c**, representative images are shown. Arrowheads indicate p-Ulk1[746] signals. In **d**, the population of cells displaying p-Ulk1[746] signals was calculated ($n \geq 100$ cells). $Atg5/RIPK3^{DKO}$, $Atg5/RIPK3^{DKO}$ + RIPK3 (WT): no treatment vs. etoposide: $p = 0.9997$, $p = 0.0028$, respectively. $Atg5/RIPK3^{DKO}$ vs. $Atg5/RIPK3^{DKO}$ + RIPK3 (WT): etoposide: $p = 0.0099$, $Atg5/RIPK3^{DKO}$ + RIPK3 (WT) vs. $Atg5/RIPK3^{DKO}$ + RIPK3 (KD): etoposide: $p = 0.0041$. In **e, f**, similar experiments to Fig. 2b were performed. Arrowheads indicate mRFP–GFP red puncta. In **f**, the indicated MEFs were treated with or without 10 μM of etoposide for 15 h. The extent of red puncta was indicated by the RFP/GFP ratio per cell ($n = 20$ cells). Red lines indicate the mean value. $Atg5/RIPK3^{DKO}$ + RIPK3 (KD) no treatment vs. etoposide: $p = 0.9376$, $Atg5/RIPK3^{DKO}$ vs. $Atg5/RIPK3^{DKO}$ + RIPK3 (WT), etoposide: $p = 0.0287$. **g, h** $Atg5^{KO}$ MEFs were treated with etoposide (10 μM) for 12 h with or without GSK'872 (10 μM), and immunostained with anti-p-Ulk1[746] and anti-GS28 antibodies. In **g**, representative images are shown. p-Ulk1[746] signals were abolished by the addition of GSK'872. In **h**, the population of cells displaying p-Ulk1[746] signals was calculated ($n \geq 100$ cells). $Atg5^{KO}$, $Atg5^{KO}$ GSK'872: no treatment vs. etoposide: $p = 0.0164$, $p = 0.7853$, respectively. $Atg5^{KO}$ vs. $Atg5^{KO}$ GSK'872: etoposide: $p = 0.0476$. In **d** and **h**, data are shown as the mean ± SD ($n = 3$). Comparisons were performed using one-way ANOVA followed by the Tukey post-hoc test. *$p < 0.05$, **$p < 0.01$. NS: not significant. Source data are provided as a Source Data file.

(Supplementary Fig. 21b, c). Neither the phosphorylation of Ulk1 Ser[746] nor the induction of alternative autophagy were observed in $Atg5/p53^{DKO}$ MEFs was already shown (Fig. 7f–i). Collectively, upon etoposide treatment, Ulk1 at Ser[637] is dephosphorylated by the p53–PPM1D axis and then phosphorylated by the p53–RIPK3 axis. Thus, p53 plays a dual role in the phosphorylation of Ulk1 at Ser[746].

We further analyzed where and how RIPK3 phosphorylates Ulk1. To this end, we visualized the interaction between RIPK3 and Ulk1 in $Atg5^{KO}$ cells using a close proximity (Duolink) assay. As indicated, signals of RIPK3–Ulk1 interaction were increased upon etoposide treatment. Importantly, all signals were localized in the cytosol but not in the Golgi (Fig. 8c, d), indicating that the RIPK3–Ulk1 interaction occurs in the cytosol. Furthermore, other Duolink assays between HA-Ulk1 and GS28, a pan Golgi marker, showed that etoposide increases the number of HA-Ulk1 (WT)-GS28 signals in $Atg5/Ulk1^{DKO}$ MEFs (Fig. 8e, f). The signals were largely decreased when analyzed between HA-Ulk1 (S746A) and GS28 (Fig. 8e, f), indicating that RIPK3-dependent Ser[746] phosphorylation is important for the translocation of Ulk1 to the Golgi (Fig. 8g).

Fip200 and Atg13 are known as binding partners of Ulk1 in canonical autophagy[2,3]. However, their interaction was dramatically decreased in etoposide-treated $Atg5^{KO}$ MEFs (Fig. 8h, lanes 7–9), and this decrease was not observed, but was rather increased, in $Atg5/RIPK3^{DKO}$ cells (Fig. 8h, lanes 10–12), indicating that Ulk1 phosphorylation at Ser[746] and Golgi translocation facilitate dissociation of the Ulk1/Fip200/Atg13 complex during alternative autophagy. Consistently, when we analyzed the interaction of HA-Ulk1 (S746D) and HA-Ulk1 (S746A) with Atg13/Fip200 in $Atg5/Ulk1^{DKO}$ MEFs upon etoposide treatment, we observed a positive interaction for HA-Ulk1 (S746A), but not for HA-Ulk1 (S746D) (Supplementary Fig. 22a). Furthermore, Fip200 and Atg13 were not localized to the Golgi in etoposide-treated $Atg5^{KO}$ MEFs (Supplementary Fig. 22b, c). Therefore, Ulk1 dissociation from the Ulk1/Fip200/Atg13 complex may be important for the recruitment of other target substrates to the Golgi. Taken together, RIPK3 phosphorylates dephospho-Ulk1[637] in the cytosol, and p-Ulk1[746] translocates to the Golgi (Fig. 8g).

**Biological roles of etoposide-induced alternative autophagy.** We finally addressed the biological roles of etoposide-induced alternative autophagy. First, we analyzed the effects of alternative autophagy on apoptosis, because we previously

showed that the degradation of Noxa inhibits genotoxic stress-induced apoptosis by canonical autophagy[14]. However, genotoxic stress-induced thymocyte cell death was not altered by RIPK3 (Supplementary Fig. 23a, b), and Noxa expression levels were also unaffected (Supplementary Fig. 23c). Similar results were obtained when using splenocytes (Supplementary Fig. 23d, e). Because RIPK3 deficiency blocks alternative autophagy, these results demonstrate that apoptosis may not be regulated by alternative autophagy, at least in thymocytes and splenocytes.

We next focused on Golgi trafficking, because we previously showed that alternative autophagy is induced by the disturbance of Golgi-mediated trafficking for the degradation of undelivered cargos[8]. Thus, we suspected that etoposide also disturbs Golgi trafficking and that alternative autophagy degrades undelivered Golgi cargos. Consistently, genotoxic stress-induced alterations of Golgi morphology has already been reported[24]. To investigate the association between genotoxic stress and Golgi trafficking, we expressed the vesicular stomatitis virus ts045 G protein fused to GFP (VSVG–GFP) in $Atg5^{KO}$ MEFs[8], and analyzed their location and degradation (Fig. 9a). VSVG–GFP was synthesized and remained in the ER at the restrictive temperature (40 °C) (Fig. 9b, image 1). By shifting to the permissive temperature (32 °C), it was transported to the Golgi for 15 min, which was confirmed by its merge with the Golgi marker GS28 (Fig. 9b, image 2), and was delivered to the plasma membrane (PM) in 60 min (Fig. 9b, image 3). However, etoposide treatment suppressed this delivery, because of the low expression of PM-localized VSVG–GFP even at 60 min after the temperature shift (Fig. 9b, image 6). The low expression of PM-localized VSVG–GFP was confirmed using flow cytometry (Fig. 9c). Interestingly, we observed some VSVG–GFP not only in the Golgi, but also in lysosomes (as assessed by Lamp2 staining), which was substantially increased by the addition of E64d/pepstatin, broad inhibitors of lysosomal proteases (Fig. 9d arrows). Quantitative analysis confirmed the increase in cellular VSVG–GFP levels by E64d/pepstatin (Fig. 9e). Because E64d/pepstatin inhibits intralysosomal degradation, this indicates that alternative autophagy is utilized for the degradation of untransported VSVG–GFP upon etoposide treatment. In contrast to $Atg5^{KO}$ MEFs, in $Atg5/RIPK3^{DKO}$ MEFs, VSVG–GFP was observed not only on the Golgi but also in the cytoplasmic punctate structures at 60 min (Fig. 9b, compare images 6 and 12). A higher expression level of VSVG–GFP in $Atg5/RIPK3^{DKO}$ MEFs than that in $Atg5^{KO}$ MEFs was quantitatively confirmed (Fig. 9e). Furthermore, E64d/pepstatin did not show any effects on VSVG–GFP localization (Fig. 9d) and expression levels (Fig. 9e), and

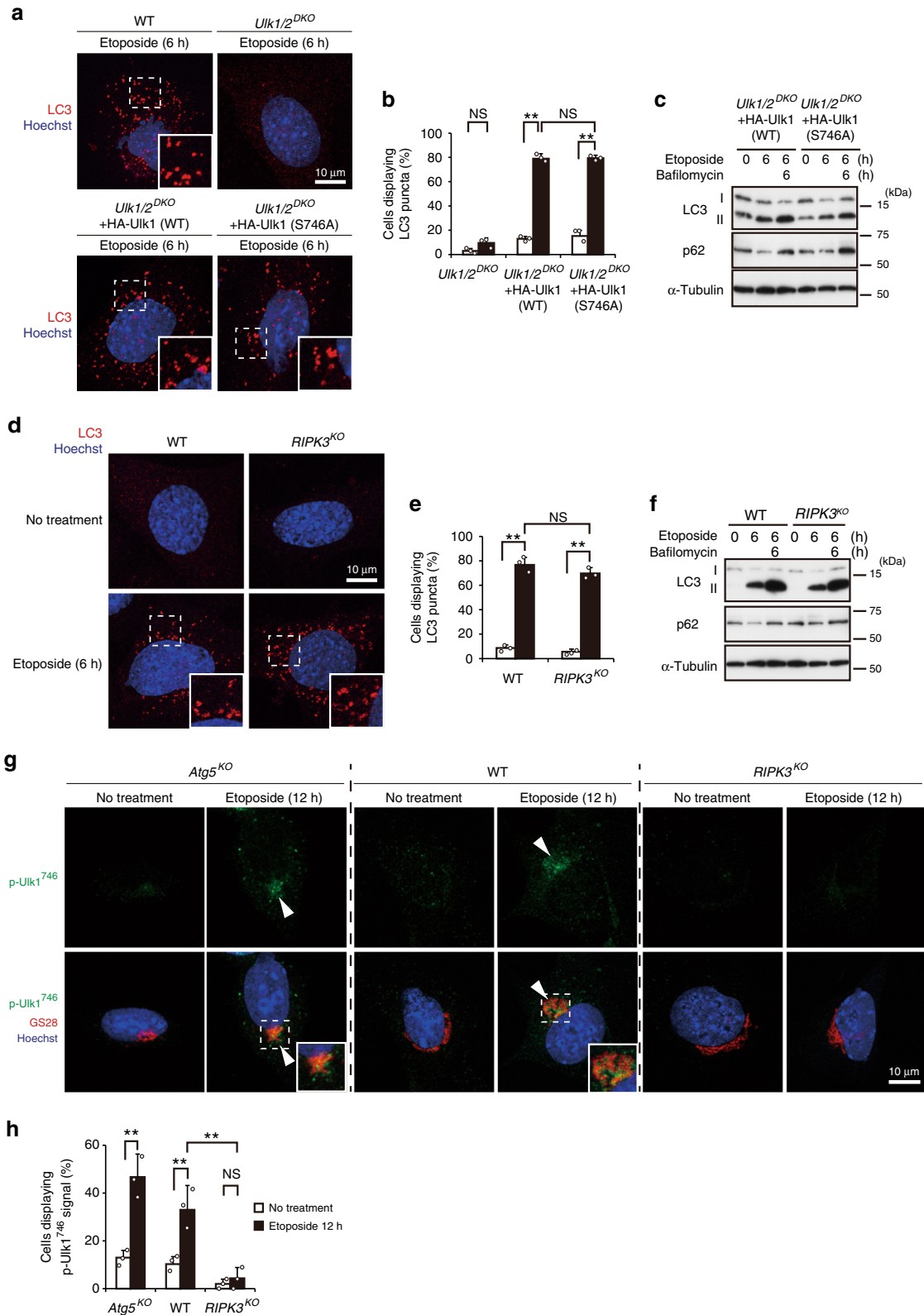

colocalization of VSVG–GFP and Lamp2 was only rarely observed in *Atg5/RIPK3^DKO* MEFs, which indicates that VSVG–GFP was not engulfed by autolysosomes. Therefore, RIPK3-dependent alternative autophagy is thought to play a role in eliminating superfluous undelivered proteins from the Golgi in etoposide-treated MEFs. Consistent results were

obtained when cells were treated with camptothesin (Supplementary Fig. 24). We also performed the VSVG–GFP trafficking assay using *Atg5/Ulk1^DKO* MEF derivatives and found that HA-Ulk1 (WT)-expressing *Atg5/Ulk1^DKO* MEFs and HA-Ulk1 (S746A)-expressing *Atg5/Ulk1^DKO* MEFs showed similar results to *Atg5^KO* MEFs and *Atg5/RIPK3^DKO* MEFs, respectively

**Fig. 4 Ulk1 Ser[746] phosphorylation and RIPK3 are not involved in canonical autophagy. a, b** The indicated MEFs were treated with or without etoposide (10 μM) for 6 h, and immunostained with an anti-LC3 antibody. Nuclei were counterstained with Hoechst 33342. Representative images of LC3 (red) and Hoechst 33342 (blue) are shown. Magnified images of the areas within the dashed squares are shown in the inset. The population of cells with LC3 puncta **b** was calculated ($n \geq 100$ cells). Ulk1/2[DKO] no treatment vs. etoposide: $p = 0.2269$. **c** The indicated MEFs were treated with etoposide (10 μM) for 6 h in the presence or absence of bafilomycin A1 (10 nM), and the expression of each protein was analyzed by western blotting. α-Tubulin was included as a loading control. **d–f** Similar experiments as **a–c** were performed using WT and RIPK3[KO] MEFs. In **e**, WT vs. RIPK3[KO] etoposide: $p = 0.2492$. **g, h** Requirement of RIPK3 for the etoposide-induced generation of p-Ulk1[746]. Similar experiments to Fig. 1d were performed using the indicated MEFs. In **g**, representative images of p-Ulk1[746] (upper panels) and merged images (lower panels) of p-Ulk1[746], GS28, and Hoechst 33342 are shown. Arrowheads indicate p-Ulk1[746] signals. In **h**, the population of cells displaying p-Ulk1[746] signals was calculated ($n \geq 100$ cells). Data are shown as the mean ± SD ($n = 3$). Atg5[KO], WT, RIPK3[KO], no treatment vs. etoposide: $p = 0.0003$, $p = 0.0085$, $p = 0.997$, respectively. WT vs. RIPK3[KO] etoposide: $p = 0.0013$. Comparisons were performed using one-way ANOVA followed by the Tukey post-hoc test. **$p < 0.01$; NS: not significant. Source data are provided as a Source Data file.

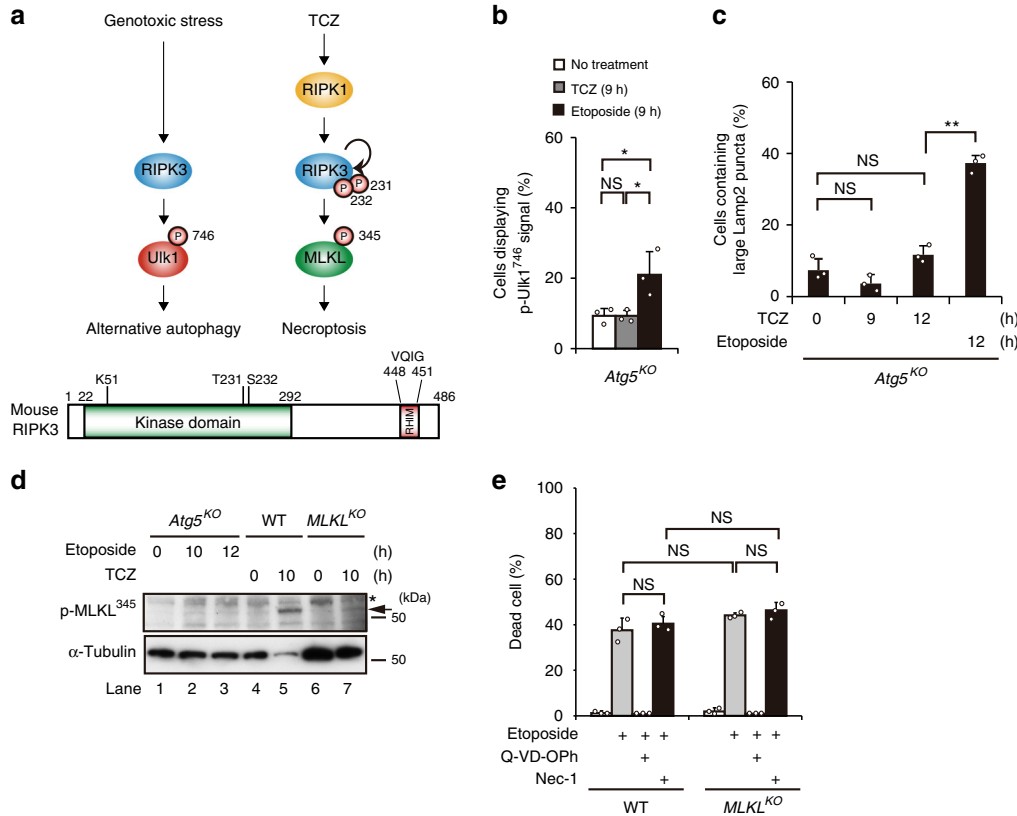

**Fig. 5 Absence of crosstalk between TCZ-induced necroptosis and etoposide-induced alternative autophagy. a** (Upper figures) Schematic model of RIPK3-dependent alternative autophagy and TCZ-induced necroptosis. Genotoxic stress induces alternative autophagy via RIPK3-dependent Ulk1 phosphorylation at Ser[746]. TCZ induces necroptosis via RIPK1, RIPK3, and MLKL. (Lower figure) Structure of RIPK3. The kinase and RHIM domains are indicated. Amino acid numbers are shown at the top. T231 and S232 are sites phosphorylated by TCZ treatment. **b, c** p-Ulk1[746] and alternative autophagy are not induced by TCZ treatment. Atg5[KO] MEFs were treated with or without the necroptosis-inducing TCZ solution for 9 h, and then immunostained with anti-p-Ulk1[746] and anti-GS28 antibodies or an anti-Lamp2 antibody. The population of cells displaying p-Ulk1[746] signals **b** and that of cells containing large Lamp2 puncta ($\geq 2$ μm) **c** was calculated ($n \geq 100$ cells). Data of cells treated with etoposide are shown as a positive control. Representative images are shown in Supplementary Fig. 14. In **b**, no treatment vs. etoposide: $p = 0.0293$, TCZ vs. etoposide: $p = 0.0293$. In **c**, TCZ 0 h vs. TCZ 9 h: $p = 0.4194$, TCZ 0 h vs. TCZ 12 h: $p = 0.2938$. **d** MLKL is not activated in etoposide-treated Atg5[KO] MEFs. The indicated MEFs were treated with etoposide (10 μM) or TCZ for the indicated times, and phosphorylated MLKL was analyzed by western blotting using an anti-p-MLKL[345] antibody. The p-MLKL signal (arrow) was observed only in TCZ-treated WT MEFs (lane 5). The asterisk indicates a nonspecific band. **e** Induction of apoptosis, but not necroptosis, by etoposide treatment. The indicated MEFs were treated with etoposide (10 μM) together with Q-VD-OPh (50 μM) or Nec-1 (30 μM) for 16 h, and the population of dead cells was analyzed by the PI uptake assay. WT etoposide vs. WT etoposide/Nec-1: $p = 0.864$, WT etoposide vs. MLKL[KO] etoposide: $p = 0.1024$, WT etoposide/Nec-1 vs. MLKL[KO] etoposide/Nec-1: $p = 0.1791$. MLKL[KO] etoposide vs. MLKL[KO] etoposide/Nec-1: $p = 0.9628$. In **b**, **c**, and **e** data are shown as the mean ± SD ($n = 3$). Comparisons were performed using one-way ANOVA followed by the Tukey post-hoc test. **$p < 0.01$; NS: not significant. Source data are provided as a Source Data file.

(Supplementary Fig. 25), indicating that p-Ulk1[746]-dependent alternative autophagy plays a role in eliminating undelivered VSVG–GFP in etoposide-treated MEFs.

Because VSVG–GFP is an artificial substrate used to monitor Golgi trafficking, we further analyzed integrin alpha5 as an example of an endogenous cargo that is trafficked by the Golgi. Integrin alpha5 mostly localized to focal adhesions in Atg5[KO] MEFs (Fig. 10a). Upon the addition of etoposide and E64d/pepstatin, we observed multiple integrin alpha5 puncta encircled by Lamp1 (Fig. 10a, image 9), which

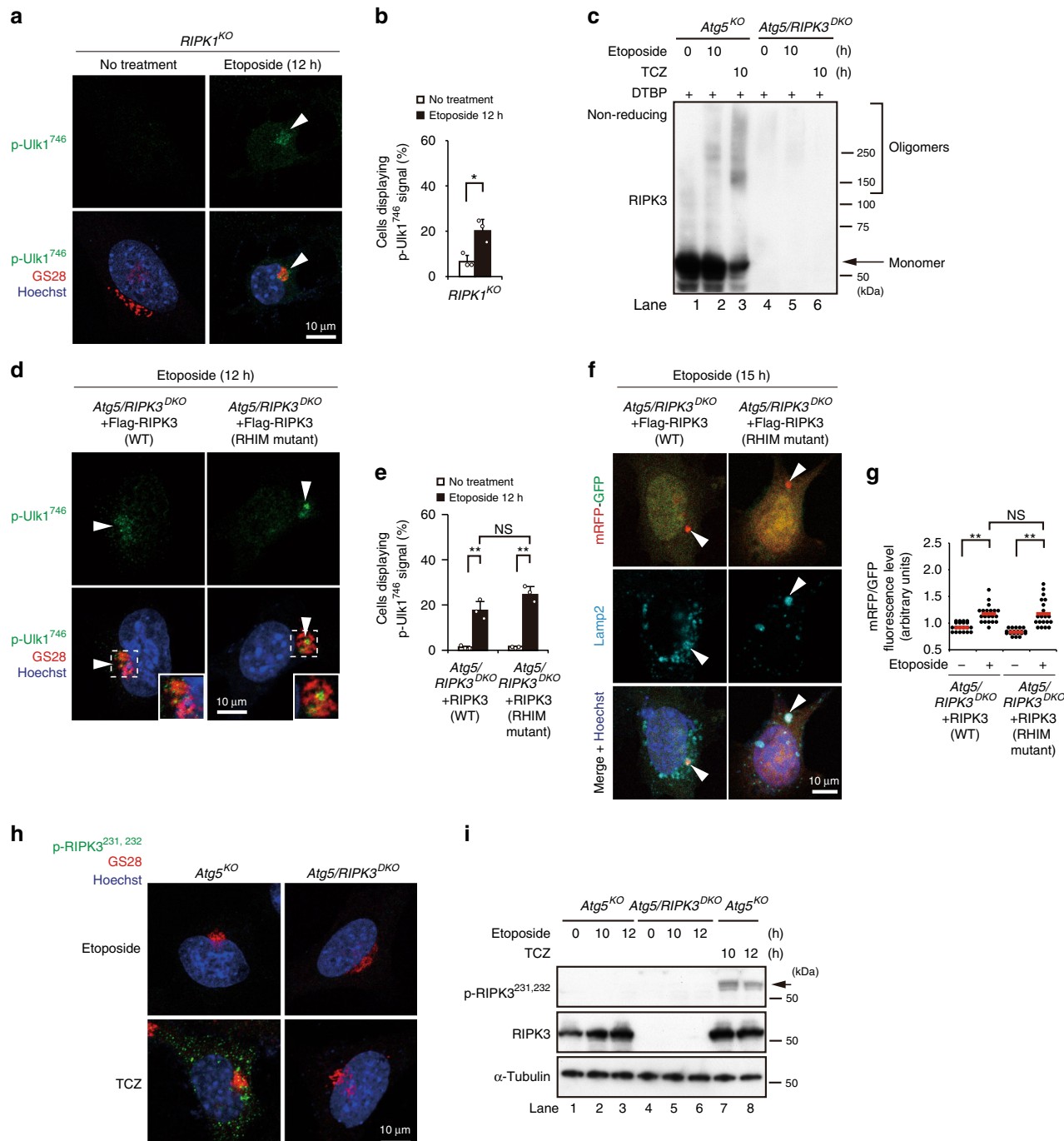

**Fig. 6 Different mechanisms of RIPK3 activation by etoposide and TCZ treatment. a, b** RIPK1 is not involved in alternative autophagy. *RIPK1^{KO}* cells were treated with or without etoposide (10 μM) for 12 h, and immunostained with anti-p-Ulk1^{746} and anti-GS28 antibodies. Arrowheads indicate p-Ulk1^{746} signals. **b** The population of cells displaying p-Ulk1^{746} signals was calculated (*n* ≥ 100 cells). *p* = 0.025. **c** Lack of RIPK3 oligomerization by etoposide treatment. The indicated MEFs were treated with or without etoposide or TCZ for 10 h. Cells were then incubated with 2 mM DTBP for 30 min. Nonreduced lysates were subjected to western blotting. Note the increase in oligomer bands and the decrease in the monomer band only by TCZ treatment. **d–g** No requirement of the RHIM domain in etoposide-induced alternative autophagy. Similar experiments to Figs. 1d and 2b were performed using Flag-RIPK3 (WT)-transfected *Atg5/RIPK3^{DKO}* MEFs and Flag-RIPK3 (RHIM mutant)-transfected *Atg5/RIPK3^{DKO}* MEFs. Representative images are shown in **d**. Magnified images are shown in the insets. Arrowheads indicate p-Ulk1^{746} signals. In **e**, the population of cells displaying p-Ulk1^{746} signals was calculated (*n* ≥ 100 cells). *Atg5/RIPK3^{DKO}* + RIPK3 (WT): no treatment vs. etoposide: *p* = 0.0004, *Atg5/RIPK3^{DKO}* + RIPK3 (WT) vs. *Atg5/RIPK3^{DKO}* + RIPK3 (RHIM mutant): etoposide: *p* = 0.0545. Representative images are shown in **f**. Arrowheads indicate mRFP–GFP red puncta. In **g**, the extent of red puncta was indicated by the RFP/GFP ratio per cell (*n* = 20 cells). Red lines indicate the mean value. *Atg5/RIPK3^{DKO}* + RIPK3 (WT) vs. *Atg5/RIPK3^{DKO}* + RIPK3 (RHIM mutant) etoposide: *p* = 0.997. **h, i** p-RIPK3^{231,232} is not induced in etoposide-induced alternative autophagy. The indicated MEFs were treated with etoposide (10 μM) or TCZ for 10 h. In **h**, cells were immunostained with anti-p-RIPK3^{231,232} and anti-GS28 antibodies. In **i**, the expression of each protein was analyzed by western blotting. The p-RIPK3^{231,232} signal (arrow) was observed in TCZ-treated MEFs, but not etoposide-treated MEFs. In **b** and **e**, data are shown as the mean ± SD (*n* = 3). Comparisons were performed using unpaired two-tailed Student *t*-tests in **b** or using one-way ANOVA followed by the Tukey post-hoc test **e** and **g**. *\*p* < 0.05; *\*\*p* < 0.01; NS: not significant. Source data are provided as a Source Data file.

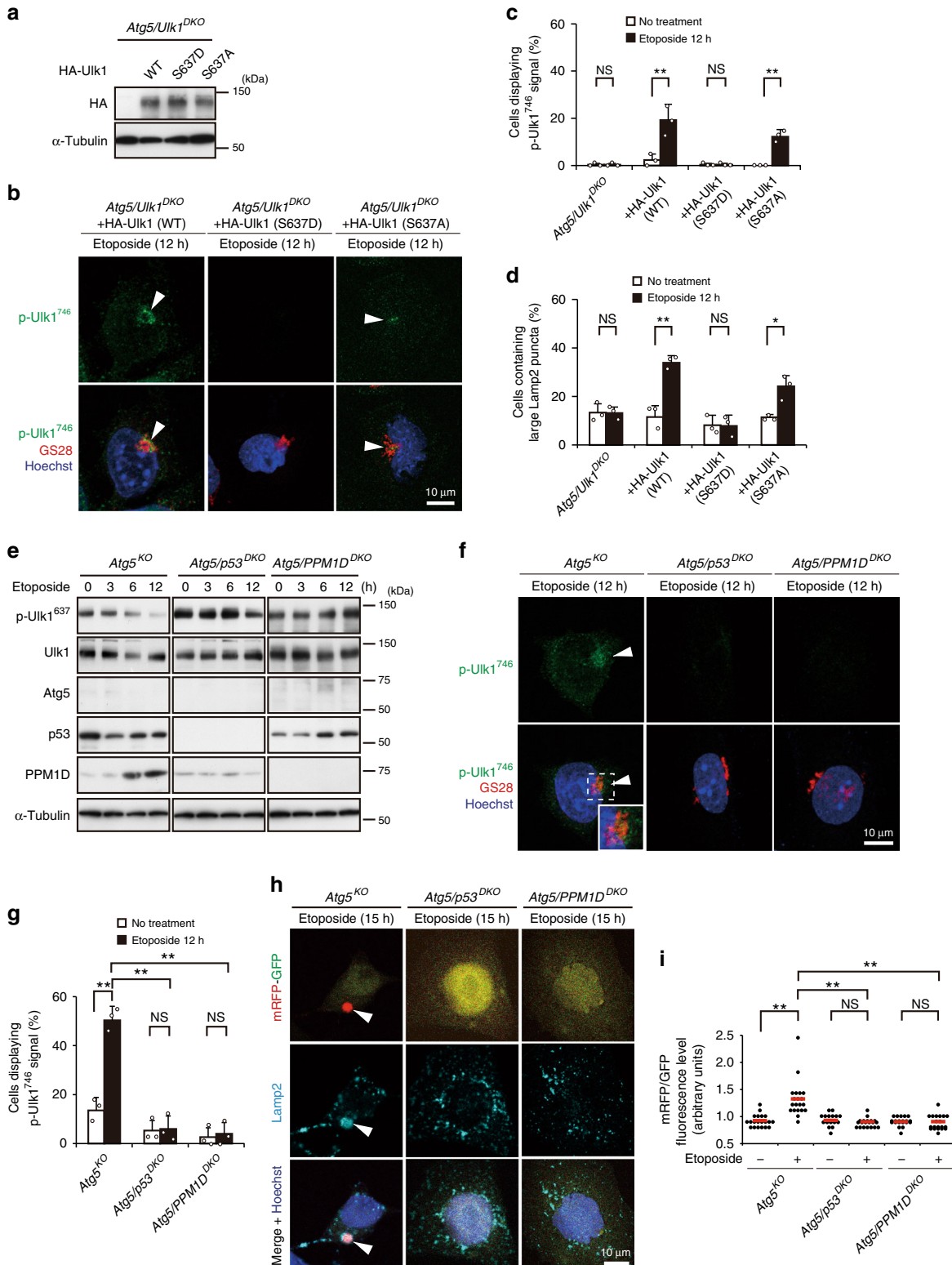

are thought to be autolysosomes. In contrast, in *Atg5/RIPK3*[DKO] MEFs, etoposide generated unusual cytoplasmic integrin alpha5 puncta (Fig. 10a, image 13), which should be the undelivered and undegraded integrin alpha5. E64d/pepstatin did not show any effects on this integrin alpha5 localization (Fig. 10a, image 16) and expression (Fig. 10b). Consistently, merged puncta of integrin alpha5 and Lamp1

were only rarely observed in *Atg5/RIPK3*[DKO] MEFs (Fig. 10a, image 18). HA-Ulk1 (WT)-expressing and HA-Ulk1 (S746A)-expressing *Atg5/Ulk1*[DKO] showed similar results with *Atg5*[KO] MEFs and *Atg5/RIPK3*[DKO] MEFs, respectively (Supplementary Fig. 26). Taken together, RIPK3/p-Ulk1[746]-dependent alternative autophagy plays a role in eliminating undelivered Golgi cargos upon genotoxic stress.

**Fig. 7 Requirement of p53/PPM1D-dependent dephosphorylation of Ulk1 at Ser$^{637}$ before the phosphorylation of Ulk1 at Ser$^{746}$. a** Expression of HA-Ulk1 mutants in *Atg5/Ulk1*$^{DKO}$ cells was confirmed by western blotting. **b, c** The indicated MEFs were treated with or without etoposide (10 μM) for 12 h, and immunostained with anti-p-Ulk1$^{746}$ and anti-GS28 antibodies. In **b**, representative images of p-Ulk1$^{746}$ and merged images are shown. Arrowheads indicate p-Ulk1$^{746}$ signals. In **c**, the population of cells displaying p-Ulk1$^{746}$ signals was calculated ($n \geq 100$ cells). *Atg5/Ulk1*$^{DKO}$ + HA-Ulk1 (S637A) no treatment vs. etoposide: $p = 0.0012$. **d** The indicated MEFs were treated with or without etoposide (10 μM) for 12 h, and immunostained with anti-Lamp2 antibody. The population of cells containing large Lamp2 puncta was calculated ($n \geq 100$ cells). Representative images are shown in Supplementary Fig. 17. *Atg5/Ulk1*$^{DKO}$ + HA-Ulk1 (S637A) no treatment vs. etoposide: $p = 0.0108$. **e** Effects of p53 and PPM1D in the etoposide-induced modification of Ulk1. The indicated MEFs were treated with etoposide (10 μM) for the indicated times, and the expression of each protein was analyzed by western blotting. **f, g** Requirement of p53 and PPM1D for the etoposide-induced phosphorylation at Ulk1 Ser$^{746}$. Similar experiments to Fig. 1d were performed. In **f**, representative images of p-Ulk1$^{746}$ and merged images are shown. Arrowheads indicate p-Ulk1$^{746}$ signals. In **g**, the population of cells displaying p-Ulk1$^{746}$ signals was calculated ($n \geq 100$ cells). p-Ulk1$^{746}$ signals were not observed in *Atg5/p53*$^{DKO}$ MEFs and *Atg5/PPM1D*$^{DKO}$ MEFs. *Atg5/PPM1D*$^{DKO}$, no treatment vs. etoposide: $p = 0.9933$. **h, i** Requirement of p53 and PPM1D for etoposide-induced alternative autophagy. Similar experiments to Fig. 2b were performed using the indicated MEFs. In **h**, representative images are shown. Arrowheads indicate mRFP–GFP red puncta. In **i**, the extent of red puncta was indicated by the RFP/GFP ratio per cell ($n = 20$ cells). Red lines indicate the mean value. *Atg5/p53*$^{DKO}$, no treatment vs. etoposide: $p = 0.9201$, *Atg5/PPM1D*$^{DKO}$, no treatment vs. etoposide: $p = 0.9987$. In **c**, **d**, and **g** data are shown as the mean ± SD ($n = 3$). Comparisons were performed using one-way ANOVA followed by the Tukey post-hoc test. **$p < 0.01$; NS: not significant. Source data are provided as a Source Data file.

## Discussion

Both canonical and alternative autophagy are initiated by Ulk1 activation, but the mechanism as to how Ulk1 activates these different pathways has remained unknown. We here demonstrated that the phosphorylation of Ulk1 at Ser$^{746}$ is the key difference between the two autophagy pathways, because the extent of Ulk1$^{746}$ phosphorylation determines the induction of alternative autophagy but not canonical autophagy. Therefore, the induction of p-Ulk1$^{746}$ should be a good marker of alternative autophagy.

Ulk1 is usually located throughout the cytosol, and upon genotoxic stress, a proportion of Ulk1 is phosphorylated at Ser$^{746}$ by RIPK3 in the cytosol. How does p-Ulk1$^{746}$ localize on the Golgi? First, genotoxic stress dephosphorylates Ulk1 at Ser$^{637}$ in a manner dependent on p53 and PPM1D. This dephosphorylation is important for both canonical and alternative autophagy. Next, p53-upregulated RIPK3 associates with and phosphorylates Ulk1 at Ser$^{746}$ in the cytosol, and subsequently p-Ulk1$^{746}$ translocates to the Golgi. It is thought that Ser$^{637}$ dephosphorylation is required for the structural alteration of Ulk1, which is required for the phosphorylation on Ulk1$^{746}$. Phosphorylated Ulk1$^{746}$ should dissociate from Fip200/Atg13, a complex required for canonical autophagy, and instead, it translocates to the Golgi and interacts with and phosphorylates yet unidentified substrates that are localized on the Golgi, to induce alternative autophagy.

We here identified RIPK3 as the upstream kinase of Ulk1 during etoposide-induced alternative autophagy. RIPK3 is also known as the upstream kinase of MLKL during TCZ-induced necroptosis[22,23]. For MLKL activation, RIPK3 recruitment by RIPK1, RHIM domain-mediated RIPK3 oligomerization, and phosphorylation of RIPK3 on Thr$^{231}$/Ser$^{232}$ are required. In contrast, for etoposide-induced Ulk1 phosphorylation, RIPK3 activation occurs independently of RIPK1, RIPK3 oligomerization, and p-RIPK3$^{231,232}$ generation. Instead, it occurs via the p53-dependent transcriptional upregulation. Then, the upregulated RIPK3 phosphorylates Ulk1 in the cytosol. Because there are several other RIPK3 target molecules, such as metabolic enzymes[25] (glycogen phosphorylase L, glutamate-ammonia ligase, and glutamate dehydrogenase 1) in the cytosol, these substrates might also be activated and play a role in various cellular events.

The major phosphorylation consensus motifs of RIPK3 are phospho-Thr/phospho-Ser/Met, phospho-Ser/Pro, and phospho-Ser/X/Pro[19,22,26]. The position of Ulk1$^{746}$ (phospho-Ser/Ser/Pro) fits with this consensus motif, whereas Ulk2 does not have any Ser/Ser/Pro or Ser/Pro sequences in the local vicinity of the

alignment locations (Supplementary Fig. 1e). Therefore, RIPK3 may only target Ulk1, but not Ulk2, upon genotoxic stress. Because *Atg5/Ulk1*$^{DKO}$ MEFs and *Atg5/RIPK3*$^{DKO}$ MEFs were largely resistant to genotoxic stress-induced alternative autophagy, the contribution of Ulk2 to genotoxic stress-induced alternative autophagy appears to be small. Of course, we do not deny the possibility that Ulk2 plays a role in alternative autophagy in other contexts. Taken together, we here identified a phosphorylation site (Ser$^{746}$) of Ulk1, which is phosphorylated upon genotoxic stress in a RIPK3-dependent manner. p-Ulk1$^{746}$ localizes solely to the Golgi, and is essential for alternative autophagy, but not canonical autophagy.

## Methods

**Mice.** *RIPK3*$^{KO}$ mice[20] and *PPM1D*$^{KO}$ mice[27] were kind gifts from Genentech Co. and Professor L.A. Donehower, respectively. Mice were bred in a 12 h light/12 h dark cycle at ~23 °C and 40% relative humidity at the Laboratory for Recombinant Animals of Tokyo Medical and Dental University, Tokyo, Japan. The Tokyo Medical and Dental University Ethics Committee for Animal Experiments approved all experiments in this study, and all experiments were performed according to their regulations.

**Antibodies and chemicals.** The antibodies used are listed in Supplementary Table 1. Etoposide, bafilomycin A1, cycloheximide, Q-VD-OPh, camptothecin, and GSK'872 were obtained from Sigma-Aldrich, Adipogen, Wako, R&D systems, Wako, and Calbiochem, respectively. All other chemicals were purchased from Nacalai Tesque.

**Plasmid construction.** The HA-tagged mouse Ulk1 plasmid was a kind gift from Professor Muramatsu (Tokyo Medical and Dental University). The introduction of point mutations into mouse Ulk1 was performed using PCR with Pfu Turbo (Agilent Technologies). The introduction of deletion mutations into mouse RIPK3 was performed using PCR with PrimeSTAR GXL (Takara). An expression vector for the galactose transferase plasmid[28] was a kind gift from Animal Resource Development & Genetic Engineering, Center for Life Science Technologies, RIKEN. All constructs were confirmed by sequence analysis. The primers used are listed in Supplementary Table 2.

**Cell culture and DNA transfection.** MEFs were generated from WT, *Atg5*$^{KO}$, *Atg5/Ulk1*$^{DKO}$, *RIPK3*$^{KO}$, *Atg5/p53*$^{DKO}$, *Atg5/PPM1D*$^{DKO}$, and *Mlkl*$^{KO}$ embryos on embryonic day 14.5 by immortalization with the SV40 T antigen. To generate MEFs stably expressing RIPK3, HA-Ulk1, and its mutants, each plasmid was transfected into MEFs ($1 \times 10^6$) using the Amaxa electroporation system (Lonza) and selected using hygromycin B (Invivogen). For transient DNA transfection, cells were transfected with the Neon electroporation system (Invitrogen) according to the manufacturer's instructions. *Atg5/RIPK3*$^{DKO}$ MEFs were generated from *RIPK3*$^{KO}$ MEFs by the CRISPR/Cas9 system[29]. Briefly, a 20-bp mouse Atg5-targeting sequence (GAGAGTCAGCTATTTGACGT) was synthesized (Eurofins) and inserted into the px330 plasmid[8] (Addgene). This plasmid was also transfected into *Ulk1/2*$^{DKO}$ MEFs. Then, a single colony of *Atg5/Ulk1/2*$^{TKO}$ and *Atg5/RIPK3*$^{DKO}$ MEFs was obtained, and Atg5 deficiency was confirmed by western blotting against Atg5 and LC3. WT and *RIPK3*$^{KO}$ primary thymocytes were

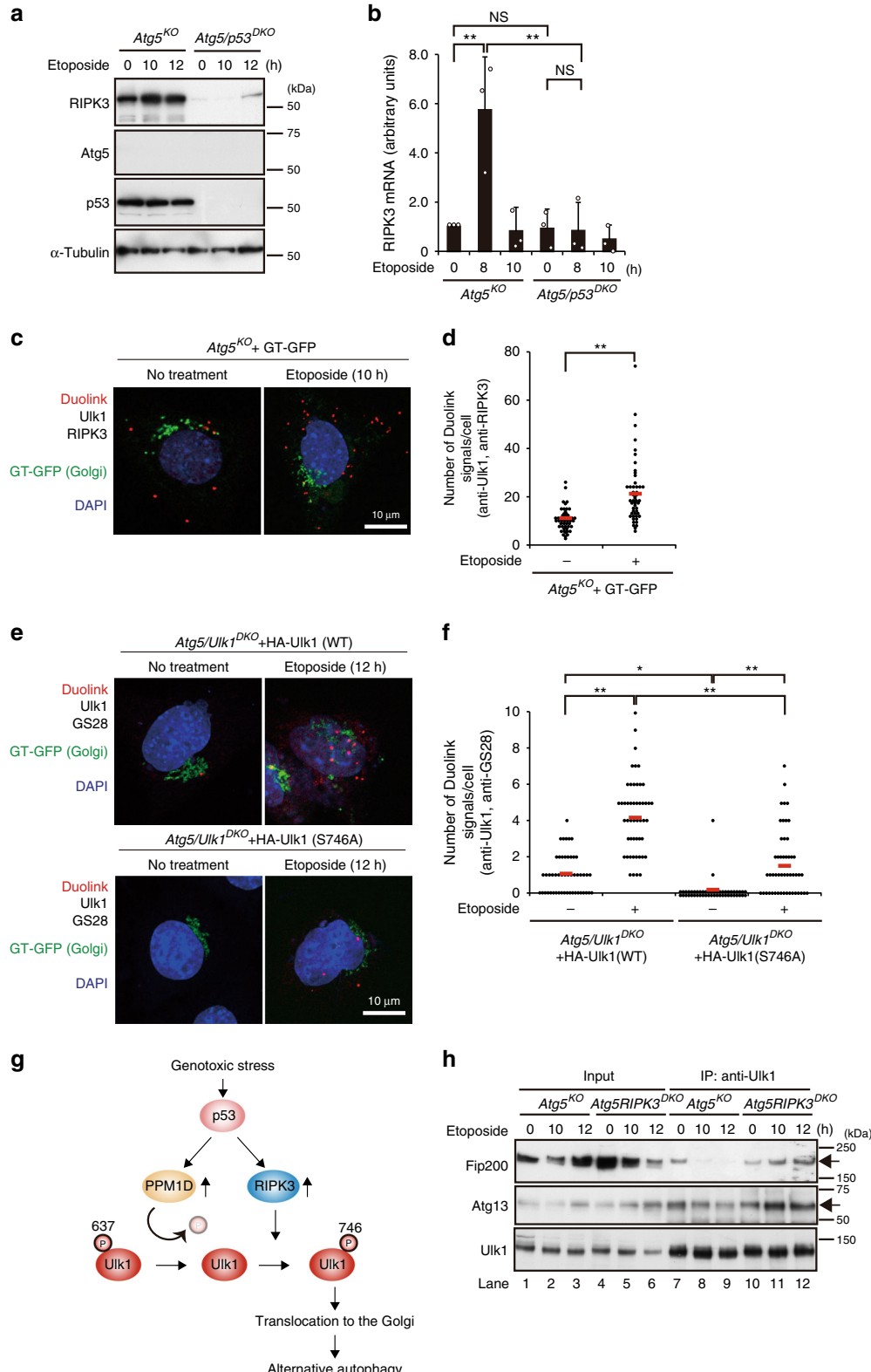

**Mass-spectrometry analysis**. Mass-spectrometry analysis was performed as follows[30]. Anti-Ulk1 immunoprecipitates from non-treated or etoposide-treated *Atg5*KO MEFs were separated by 5–20% SDS–PAGE, and stained by silver staining (SilverQuest staining kit, Invitrogen). The gel bands around 150 kDa were cut out, treated with dithiothreitol (10 mM), dissolved in ammonium hydrogen carbonate (100 mM), and treated with iodoacetamide (40 mM). After the gels were dried, 20 μL of 0.05 pmol L⁻¹ trypsin solution was applied to each gel piece and incubated

harvested from the respective mice at 5–6 weeks of age. MEFs and thymocytes were cultured in Dulbecco's modified Eagle's medium supplemented with 2 mM L-glutamine, 1 mM sodium pyruvate, 0.1 mM nonessential amino acids, 10 mM HEPES/Na⁺ (pH 7.4), 0.05 mM 2-mercaptoethanol, 100 U/mL penicillin, 100 μg/mL streptomycin, and 10% fetal bovine serum[10]. 293T cells were transfected with Lipofectamine 2000 (Thermo Fisher Scientific) according to the manufacturer's instructions.

**Fig. 8 Mechanism of Ulk1[746] phosphorylation during etoposide treatment. a, b** Crucial role of p53 in RIPK3 activation. The indicated MEFs were treated with etoposide (10 μM) for the indicated times. In **a**, the expression of each protein was analyzed by western blotting. In **b**, RIPK3 mRNA and 18S rRNA levels were analyzed using real-time PCR and their ratios were calculated. Values indicate the ratio to "$Atg5^{KO}$ time 0". Data are shown as the mean ± SD ($n = 3$). $Atg5^{KO}$, no treatment vs. etoposide: $p = 0.0029$, $Atg5^{KO}$ vs. $Atg5/p53^{DKO}$ etoposide: $p = 0.0023$. **c, d** RIPK3–Ulk1 interaction occurs in the cytosol. $Atg5^{KO}$ MEFs were transfected with an expression vector for galactose transferase-fused GFP (GT-GFP); a *trans*-Golgi marker, and treated with or without etoposide (10 μM). The physical interaction between RIPK3 and Ulk1 was analyzed using Duolink reagents. All Duolink signals were localized in the cytosol. In **d**, the number of Duolink signals were counted ($n = 50$ cells). Red bars indicate mean values. $p = 7.4 \times 10^{-7}$. **e, f** Phosphorylation-dependent translocation of Ulk1 to the Golgi. **e** The indicated MEFs were transfected with GT-GFP and treated with or without etoposide (10 μM), and assayed with Duolink reagents using anti-Ulk1/anti-GS28 antibodies. In **f**, the number of Duolink red signals were counted ($n = 50$ cells). Red bars indicate mean values. $Atg5/Ulk1^{DKO}$ + HA-Ulk1 (WT) vs. $Atg5/Ulk1^{DKO}$ + HA-Ulk1 (S746A), no treatment: $p = 0.0137$. **g** Schematic model of the RIPK3-dependent alternative autophagy. Genotoxic stress induces Ulk1 dephosphorylation at Ser[637] in a p53/PPM1D-dependent manner. The dephosphorylated Ulk1 is then phosphorylated at Ser[746] by RIPK3, and translocates to the Golgi, which results in alternative autophagy. **h** Effect of RIPK3 on the interaction of Ulk1 with Fip200 and Atg13. The indicated MEFs were treated with 10 μM of etoposide (10 μM). Cells were then lysed and immunoprecipitated with an anti-Ulk1 antibody. Immune complexes and total lysates (5.6% input) were analyzed by western blotting. Comparisons were performed using unpaired two-tailed Student *t*-tests in **d** or using one-way ANOVA followed by the Tukey post-hoc test in **b** and **f**. \*\*$p < 0.01$; NS: not significant. Source data are provided as a Source Data file.

for 14 h at 37 °C to digest the proteins. Digested peptides were extracted with 50% trifluoroacetic acid (TFA), followed by 80% TFA. The purified peptide samples were injected onto a reversed-phase C18 column (HiQ sil C18W-3P, 3 mm, 120 Å; KYA TECH Co.) and separated by nanoflow liquid chromatography (300 nL min$^{-1}$) on a nano LC Dina-A system (KYA TECH Co.) with a Q-TRAP 5500 mass spectrometer (AB SCIEX).

**Immunoblot analysis.** Cells were lysed in cell lysis buffer containing 20 mM HEPES (pH 7.5), 100 mM NaCl, 1.5 mM MgCl$_2$, 1 mM EGTA, 10 mM Na$_2$P$_2$O$_7$, 10% glycerol, 1% NP-40, 1 mM dithiothreitol, 1 mM Na$_3$VO$_4$, and 1% protease inhibitor cocktail. After vortexing for 15 s, insoluble material was removed by centrifugation. Supernatants were loaded onto 5–20%, 15%, or 8% SDS–polyacrylamide gels. After electrophoresis, the proteins were blotted onto PVDF membranes. The membranes were blocked with 5% skim milk in TBS containing 0.05% Tween-20 (TBS-T) and incubated with a primary antibody overnight at 4 °C. After washing with TBS-T, the membranes were incubated with a horseradish peroxidase-labeled secondary antibody and visualized with Chemi-Lumi One Super reagent. All experiments were conducted at least in duplicate.

**Immunoprecipitation.** Untreated and etoposide-treated MEFs were harvested and lysed with cell lysis buffer. Immunoprecipitation was performed using an indicated antibody in the presence of protein-G Sepharose (GE Healthcare) for 2 h at 4 °C. The beads were then washed three times with PBS. In experiment using Flag-RIPK3 deletion mutants in HEK293T cells, the beads were washed two times with PBS. Proteins were released from the beads by heating at 100 °C for 3 min in 2 × Laemmli sample buffer. Immunoblotting was performed as described above, except that the EasyBlot anti-rabbit IgG kit (GeneTex) was used to avoid the detection of non-specific IgG bands.

**Kinase assay.** Cells were lysed in cell lysis buffer. After vortexing for 15 s, insoluble material was removed by centrifugation. Supernatants were treated with or without 1 μg of GST-RIPK3 together with or without 2 μM of the RIPK3 inhibitor GSK'872, for 1 h at 30 °C, followed by immunoprecipitation with an anti-phospho-Ulk1[746] antibody. The levels of phosph-Ulk1[746] were analyzed by immunoblotting using an anti-Ulk1 antibody and the EasyBlot anti-rabbit IgG kit.

**Phosphatase assay.** Cells were lysed in cell lysis buffer containing 20 mM HEPES (pH 7.5), 100 mM NaCl, 1.5 mM MgCl$_2$, 1 mM EGTA, 10% glycerol, 1% NP-40, 1 mM dithiothreitol, and 1% protease inhibitor cocktail. After vortexing for 15 s, insoluble material was removed by centrifugation. Supernatants were treated with or without 4000 U of lambda protein phosphatase (λPPase) for 1 h at 30 °C, followed by immunoprecipitation with an anti-phospho-Ulk1[746] antibody. The levels of phosph-Ulk1[746] were analyzed by immunoblotting using an anti-Ulk1 antibody and the EasyBlot anti-rabbit IgG kit.

**Immunofluorescence analysis.** Cells were fixed in 4% paraformaldehyde containing 8 mM EGTA for 10 min and then permeabilized using 50 μg mL$^{-1}$ digitonin for 5 min. Cells were then stained with the indicated primary antibodies for 1 h at room temperature. After washing, the cells were stained with secondary antibodies and Hoechst 33342, mounted in Prolong Gold Antifade reagent, and observed using a fluorescence microscope (IX71, Olympus) and a laser-scanning confocal microscope (LSM710, Zeiss). For staining with the anti-Ser[746] Ulk1 antibody, cells were fixed in 4% paraformaldehyde containing 8 mM EGTA for 5 min and then permeabilized using 20 μg mL$^{-1}$ digitonin for 2 min on ice to avoid nonspecific nuclear staining. Thymocytes were fixed in 4% paraformaldehyde

containing 8 mM EGTA for 5 min, pelleted onto slides using a Cytospin3 centrifuge (Shandon), and permeabilized using 20 μg mL$^{-1}$ digitonin for 2 min on ice. Cells were then stained as described above. Data analysis was performed using Zen software 2012 (Zeiss), Adobe photoshop CS5.1, and Illustrator CS5.1.

**Quantitation of alternative autophagy.** The extent of alternative autophagy was analyzed by the mRFP–GFP assay and Lamp2 swelling assay. In the experiments using mRFP–GFP, autolysosomes were visualized as red puncta, because GFP fluorescence, but not RFP fluorescence, is decreased in acidic compartments. Therefore, the cellular RFP/GFP fluorescence ratio is increased in autophagy-inducing cells. RFP and GFP images were acquired using Zen software, and analyzed cellular fluorescence intensity (intensity × area per cell) using Image J software. Then, the cellular RFP/GFP level of each cell was calculated.

Levels of alternate autophagy were also quantified using the Lamp2 staining assay. As our previous study demonstrated that large Lamp2 puncta (≥2 μm) were identical to autophagic vacuoles[5], and there were very few such big Lamp2 puncta in non-autophagic cells, cells with more than one large LAMP2 (≥2 μm) punctum was regarded as an autophagic cell, and cells were analyzed using Zen software (Zeiss). Images of Lamp2-stained cells were acquired and the number of cells with large Lamp2 puncta were calculated.

**Correlative light and electron microscopy.** To merge photographs from confocal fluorescence microscopy and EM, samples were quick frozen and transferred to 0.00001% OsO$_4$ and 0.01% glutaraldehyde in acetone (80 °C for 24 h, −30 °C for 6 h, −15 °C for 1 h, and 37 °C for 30 min), and washed with acetone. After replacing the acetone to PBS(−), the samples were visualized by confocal microscopy. Samples were then fixed in Karnovsky solution (1.5% paraformaldehyde and 3% glutaraldehyde in phosphate buffer) for 15 min at room temperature and 1% OsO$_4$ at 4 °C for 5 min. After dehydration, the samples were embedded in Epon, and thin sections were observed with a JEM 1010 electron microscope at 80 kV.

**UVC irradiation.** After replacing the medium with PBS, cells were exposed to 30 J m$^{-2}$ of UVC light in a UV crosslinker CX-2000 equipped with a UV-lamp. After irradiation, PBS was aspirated and replaced with the culture medium. After cells were incubated for 1 and 3 h, cell were fixed and stained as described above.

**Duolink in situ proximity ligation assay.** Cells were fixed in 4% paraformaldehyde containing 8 mM EGTA for 10 min and then permeabilized using 50 μg mL$^{-1}$ digitonin for 5 min. Cells were then stained with the indicated primary antibodies overnight at 4 °C. After washing, the cells were assayed with Duolink in situ reagents according to the manufacturer's instructions, and mounted in Prolong Gold Antifade reagent with DAPI (Thermo Fisher Scientific) and a laser-scanning confocal microscope (LSM710, Zeiss). Data analysis was performed using Zen and Image J software.

**Cell viability assay.** Cells were stained with propidium iodide (PI) and cell viability was detected by flow cytometry (BD; FACS Canto II). Cell populations were separated from cellular debris using FSC/SSC (Supplementary Fig. 27). Data analysis was performed using BD FACSDiva and FlowJo software.

**Real-time PCR.** Total RNA was extracted using the Qiagen RNeasy Mini kit. The synthesis of first-strand cDNA and real-time PCR was performed in the CFX96 Real-Time system by using the iTaq universal SYBR green one-step kit (Bio-Rad). Data were collected by CFX manager 3.1 (Bio-Rad). RIPK3 and 18S rRNA primers

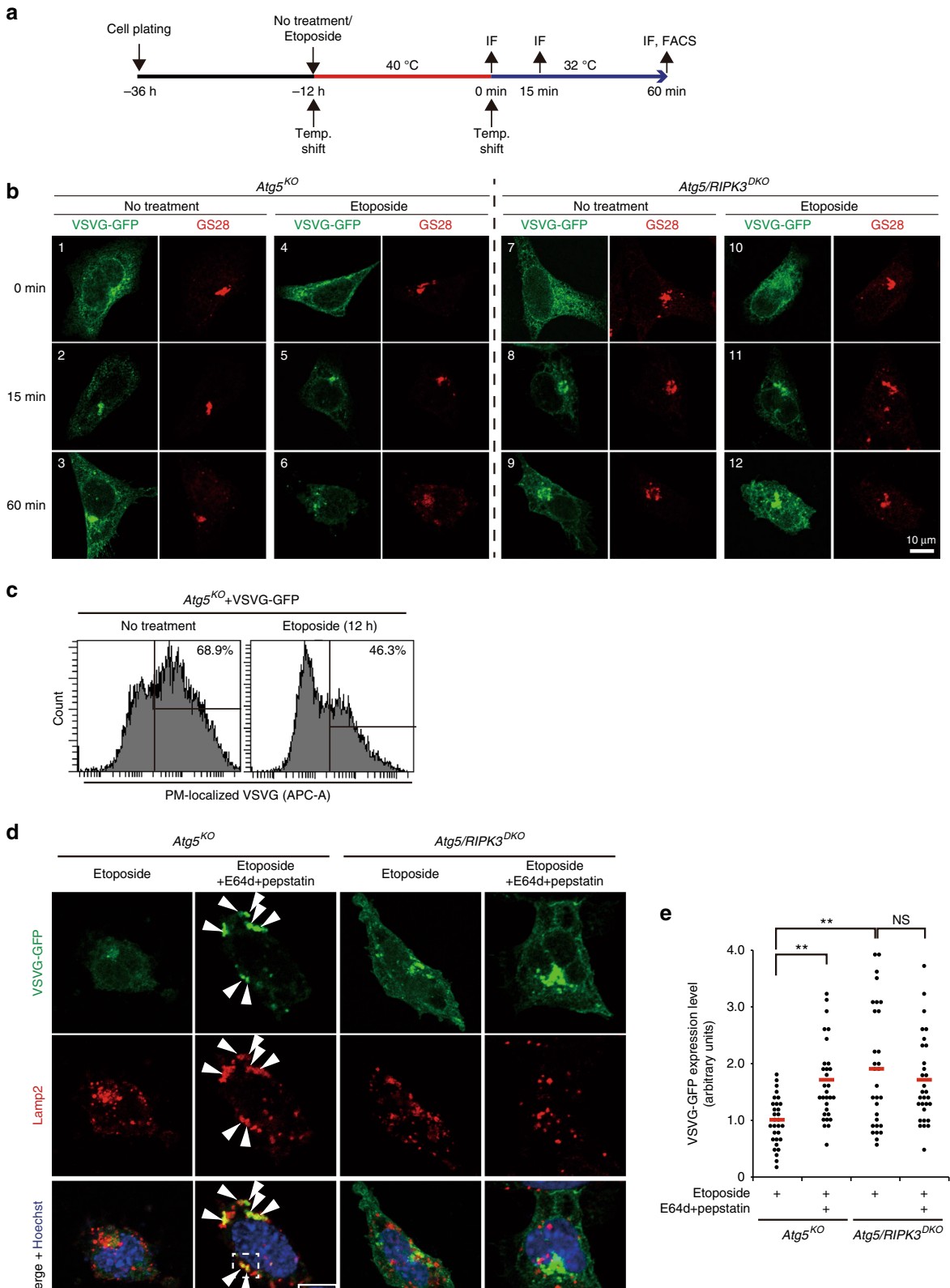

are listed in Supplementary Table 2. Expression was normalized using 18S rRNA as an internal control.

**VSVG and integrin alpha5 trafficking**. MEFs were transfected with VSVG–GFP and plated, and then treated with or without etoposide (10 μM), E64d

(10 μg mL⁻¹), and pepstatin (10 μg mL⁻¹) at the restrictive temperature for VSVG–GFP (40 °C) for 12 h. Cells were then shifted to the permissive temperature (32 °C), and fixed at the indicated times. Golgi and lysosomes were counterstained with an anti-GS28 and anti-Lamp2 antibody, respectively. Fluorescence intensity (intensity × area) per cell was calculated using Image J software. In the integrin alpha5 trafficking assay, MEFs were treated with or without etoposide (10 μM),

**Fig. 9 Effect of RIPK3 on the degradation of undelivered VSVG upon etoposide treatment. a** Schematic diagram of the experiment. The VSVG–GFP was expressed in *Atg5^KO* MEFs and *Atg5/RIPK3^DKO* MEFs, and the cells were treated with or without etoposide (10 μM) at the restrictive temperature (40 °C) for 12 h. Cells were then shifted to the permissive temperature (32 °C), and fixed at the indicate times. Golgi was counterstained with an anti-GS28 antibody. **b** Representative images are shown. At 0, 15, and 60 min after the temperature shift, the location of VSVG–GFP and Golgi were analyzed. **c** Reduction of PM-localized VSVG–GFP upon etoposide treatment. VSVG–GFP-expressing *Atg5^KO* MEFs were or were not treated with etoposide, and at 60 min after the temperature shift, cells were stained with an anti-VSVG antibody for 30 min at 4 °C followed by the addition of an Alexa Fluor 633-conjugated secondary antibody for 30 min at 4 °C. After washing the cells, the amount of PM-localized VSVG–GFP was measured using flow cytometry. **d**, **e** VSVG–GFP-expressing *Atg5^KO* MEFs and *Atg5/RIPK3^DKO* MEFs were treated with etoposide (10 μM) in the presence of E64d/pepstatin. At 60 min after the temperature shift, cells were immunostained with an anti-Lamp2 antibody. In **d**, representative images are shown. Arrowheads indicate autolysosomes containing VSVG–GFP. A magnified image of the areas within the dashed squares is shown in the insets. In **e**, the amount of total VSVG–GFP (the level of fluorescence intensity in each cell) was measured using Image J ($n = 30$ cells in each experiment). Red bars indicate mean values. *Atg5^KO* Etoposide vs. Etoposide with E64d/pepstatin: $p = 0.0043$, *Atg5/RIPK3^DKO* Etoposide vs. Etoposide with E64d/pepstatin: $p = 0.6495$. Comparisons were performed using one-way ANOVA followed by the Tukey post-hoc test. **$p < 0.01$; NS: not significant. Source data are provided as a Source Data file.

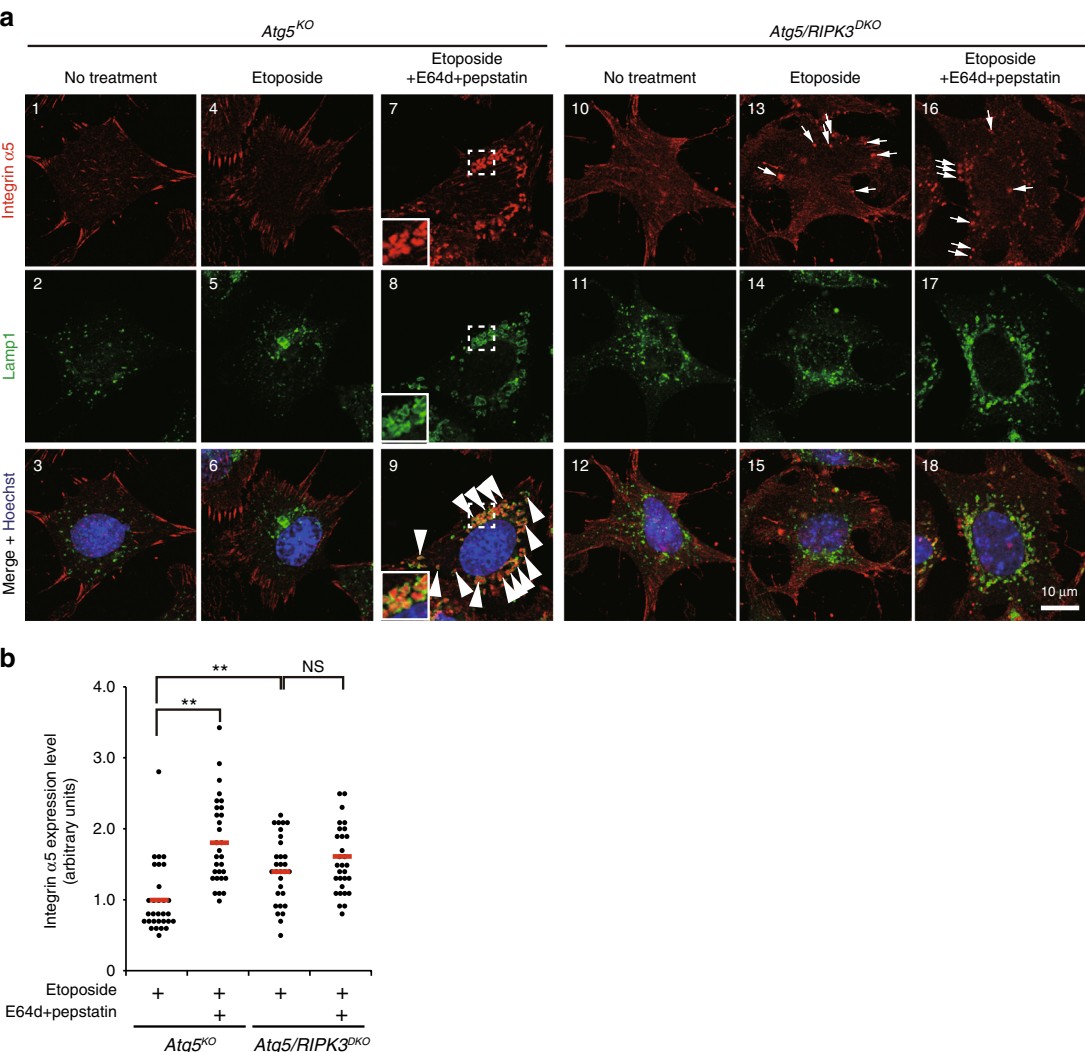

**Fig. 10 Requirement of RIPK3 for undelivered integrin α5 degradation upon etoposide treatment.** The *Atg5^KO* MEFs and *Atg5/RIPK3^DKO* MEFs were treated with or without etoposide (10 μM) in the presence of E64d/pepstatin for 12 h. Cells were immunostained with anti-Lamp1 and anti-integrin α5 antibodies. In **a**, representative images are shown. Arrows and arrowheads indicate unusual cytoplasmic integrin alpha5 puncta and integrin α5 engulfed in autolysosomes, respectively. Magnified images of the areas within the dashed squares are shown in the insets. Note that etoposide generated unusual cytoplasmic integrin alpha5 puncta in *Atg5/RIPK3^DKO* MEFs. In **b**, the amount of total integrin α5 (the level of fluorescence intensity per cell) was measured ($n = 30$ cells in each experiment). Red bars indicate mean values. *Atg5^KO* vs. *Atg5/RIPK3^DKO* Etoposide: $p = 0.0093$, *Atg5/RIPK3^DKO* Etoposide vs. Etoposide with E64d/pepstatin: $p = 0.744$. Comparisons were performed using one-way ANOVA followed by the Tukey post-hoc test. **$p < 0.01$; NS: not significant. Source data are provided as a Source Data file.

E64d (10 µg mL$^{-1}$), and pepstatin (10 µg mL$^{-1}$) for 12 h. Cells were then fixed for 12 h. Lysosomes were counterstained with an anti-Lamp1 antibody, and fluorescence intensity (intensity × area) per cell was calculated using Image J software.

For FACS analysis, cells were transfected with VSVG–GFP, treated with or without etoposide (10 µM) in the presence of Q-VD-OPh (50 µM) at the permissive temperature (32 °C) for 60 min. Cells were then stained with an anti-VSVG antibody for 30 min at 4 °C without membrane permeabilization. After washing, the cells were stained with an Alexa fluor 633-conjugated secondary antibody for 30 min. After washing, cells were analyzed by flow cytometry. Data analysis was performed using BD FACSDiva and FlowJo software.

**Statistical analysis**. Results are expressed as the mean ± standard deviation (SD). Statistical analyses were performed using Excel and Prism 8 (GraphPad) software. Comparisons of two datasets were performed using unpaired two-tailed Student $t$-tests in Figs. 6b, 8d, and Supplementary Fig. 22c. All other comparisons of multiple datasets were performed using one-way ANOVA followed by the Tukey post-hoc test. A $p$-value of <0.05 was considered to indicate a statistically significant difference between two groups. Exact $p$ values are described in figure legends, except that the values, using one-way ANOVA followed by the Tukey post-hoc test, are too large ($p > 0.9999$) or small ($p < 0.0001$)

**Statistics and reproducibility**. Repeated independent experiments per each panel with similar results are shown below. $n = 1$ (Figs. 1a, 2a and Supplementary Figs. 1c, d, and 5). $n = 2$ (Figs. 1b, c, 3a, b, 4c, f, 5d, 6c, h, i, 7a, e, 8a, c–f, h and 9c, Supplementary Figs. 1a, b, 2a, 4b, 8a, 10a–d, 12a–d, 16b, 21a, 22a–c, 23c). $n = 3$ (Figs. 1d–f, 2b–g, 3c–h, 4a, b, d, e, g, h, 5b, c, e, 6a–g, 7b–d, f–i, 8b, 9b, d, e, 10a, b and Supplementary Figs. 3a, b, 6, 7a, b, 8b–f, 9a–c, 11a–d, 13a, b, 14a, b, 15a, b, 17a, 18a, b, 19a–d, 20a, b, 21b, 23a, b, d, e, 24a, b, 25a, b and 26a–d).

**Reporting summary**. Further information on research design is available in the Nature Research Reporting Summary linked to this article.

## Data availability

All data that supporting the findings of this study are available from the corresponding author upon request. The source data underlying Figs. 1–10 and Supplementary Figs. 1–26 are provided as a Source Data file.

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

## Acknowledgements

*Atg5^{KO}*, *PPM1D^{KO}*, and *RIPK3^{KO}* mice were kindly provided by Professors N. Mizushima (University of Tokyo), L.A. Donehower (Baylor College of Medicine), and Genentech Co., respectively. *MLKL^{KO}* MEFs and *Ulk1/Ulk2^{DKO}* MEFs were kindly provided by Professors M. Pasparakis (University of Cologne) and S. Tooze (The Francis Crick Institute, UK), respectively. This study was supported in part by a Grant-in-Aid for Scientific Research (A) (17H01533), Grant-in-Aid for Scientific Research (C) (18K06210), Grant-in-Aid for challenging Exploratory Research (16K15230), Grant-in-Aid for Scientific Research on Innovative Areas (17H05691, 17H06413, 17H06414), Grant-in-Aid for Encouragement of Young Scientists (B) (15K19004) from the MEXT of Japan, by the Project for Cancer Research and Therapeutic Evolution (P-CREATE) (JP18cm0106109), by the Project for Psychiatric and Neurological Disorders (JP18dm0107136), by the Project for Practical Research Project for Rare/Intractable Diseases (JP19ek0109407), by the Translational Research program; Strategic Promotion for practical application of Innovative medical Technology, TR-SPRINT, and by the AMED-CREST (JP19gm1210002) from the Japan Agency for Medical Research and development. This study was also supported by the Joint Usage/Research Program of Medical Research Institute, Tokyo Medical and Dental University, and by a grant from the Takeda Science Foundation.

## Author contributions

S.T. designed the research, performed the biological analyses, and wrote the manuscript. A.N. performed the LC–MS/MS analyses. H.Y. and S.A. performed the CLEM analyses, S.H. provided advice regarding data interpretation, K.M. made the plasmids of RIPK3 mutants as well as advice regarding data interpretation, H.N. provided the RIPK3-deficient cells as well as advice regarding data interpretation. S.S. designed the research and wrote the manuscript. All authors edited the manuscript.

## Competing interests

The authors declare no competing interests.
