## [Peer Review File · Nature Communications]

Reviewers' comments:

Reviewer #1 (Remarks to the Author):

In the manuscript by Torii et al., the authors identify a RIPK3-dependent phosphorylation site on ULK1 that is essential for ULK's ability to induce "alternate" autophagy at the Golgi. To do this, the authors develop a phospho-specific antibody and use KO lines and rescues with WT and phosphomutant ULK1, with alternate autophagy being measured by the presence of large LAMP structures or the lysosomal delivery and selective quenching of RFP-GFP. They also find that a prior dephosphorylation event, previously discovered by the group, is a prerequisite for RIPK3-ULK1 phosphorylation and that the mechanism of RIPK3 activation is distinct from the known necroptosis pathway. Finally, the authors present evidence that the alternate autophagy pathway is important for clearance of accumulated proteins that arise due to a block in Golgi export. On the whole the experiments are well executed and the data clear. I feel that this is an important piece of work in a pathway where very little is known mechanistically. However, as the paper stands, many questions still remain (e.g. how is RIPK3 activated/ULK recruited to Golgi/consequences of this pathway etc.) and so this work represents more of an incremental step in answering these questions rather than a significant leap.

Main points

1) I am a little confused about the reasons for this alternate pathway and the consequences to a cell upon its loss. The authors uncover a very interesting cargo, related to Golgi trafficking, but what happens to the cell if this is blocked? Presumably, etoposide will eventually cause the cells to undergo apoptosis – but does loss of this alternate autophagy speed-up or delay this process? In their previous work (Torii et al., 2016 EMBOR) they proposed that loss of this (or a similar) pathway accelerated apoptosis due to reduced Noxa turnover – does loss of ULK phosphorylation have the same effect? Is the "Golgi trafficking defect" autophagy related to Noxa turnover or something different?

2) Related to the above point, the authors have developed an elegant way to disrupt this pathway through ULK KO and then rescue with WT or phospho-mutant ULK1. Why did they not use this approach in the VSVG trafficking experiments? The authors instead use RIPK3 KO, and while this blocks alternate autophagy, it will/could also have other consequences (which could in turn be important for the trafficking).

3) Also related, it was not clear how the authors decided to look at Golgi trafficking. On page 14 the authors state that DNA damage generates excess proteins by disruption of Golgi to PM trafficking, and they postulate that autophagy may be clearing this build-up. But, where does these data come from that DNA damage blocks Golgi trafficking, as there are no references? Is it related to previously published GOLPH3 studies? If so, this is an obvious candidate for the authors to analyse. Additionally, if it is DNA damage per se that is causing this pathway, then do the authors see the same phenomenon if another DNA-damaging agent is used (I do note that all the experiments here are done with etoposide)?

Other points

4) From the methods, a little more detail on how the quantitation of alternate autophagy was carried out would be helpful. One method uses cells expressing mRFP-GFP and data is displayed as cells displaying red puncta. Does this mean cells with one puncta are counted the same as cells with 10 or more puncta? In other figures, alternate autophagy is assessed by number of cells displaying large LAMP2

Reviewer #2 (Remarks to the Author):

In the manuscript by Torii et al., the authors report that RIPK3-dependent phosphorylation of ULK1 at S746 is required for genotoxic stress-induced alternative autophagy. The group previously identified ATG5-independent autophagy. Here, they report that etoposide treatment results in ULK1 phosphorylation at S746. ULK1 phosphorylated at this site apparently localizes at the Golgi. The

authors also report that S746 phosphorylation is only required for ATG5-independent autophagy, but not for canonical autophagy. Furthermore, the authors identify the necroptosis-mediating kinase RIPK3 as the kinase responsible for this phosphorylation. Finally, the authors report that p53/PPM1D-dependent dephosphorylation of ULK1 S637 is a prerequisite for RIPK3-dependent ULK1 phosphorylation. Although I think the authors made an interesting observation, I definitely think that this manuscript needs substantial revision in order to be acceptable for publication in NATURE COMMUNICATIONS. Especially, central mechanistic details remain unclear.

My comments are as follows:

Major points:

1) The authors largely focus on the RIPK3-dependent phosphorylation of ULK1 at S746. Although this represents an interesting observation, mechanistic details upstream and downstream of this phosphorylation event remain entirely unclear. The authors report the etoposide-induced activation of RIPK3 does not resemble RIPK3 activation during necroptosis, i.e. it is independent of RIPK1, oligomerization or phosphorylation of RIPK3. But then, how is RIPK3 activated? Are there any kinases involved that become activated upon etoposide-induced DNA-damage? With regard to ULK1, how does phosphorylation of S746 initiate ULK1 translocation to the Golgi? Or is this phosphorylation occurring at the Golgi? In suppl. Figure 4, it appears that the S746A-mutant can be readily transferred to the Golgi. Finally, does the phosphorylation of ULK1 at S746 alter its kinase activity or the interaction with known binding partners?

2) In figure 8, the authors try to establish the biological relevance of ATG5-independent autophagy. They suggest that ULK1 phosphorylation at S746 is important for the elimination of superfluous undelivered proteins from the Golgi. Although I agree that VSVG-GFP might co-localize with lysosomes and that VSVG-GFP expression is increased by E64d/Pepstatin, I think it is difficult to draw any conclusions from the VSVG trafficking assay. For example, for image 6 (Figure 8B) the authors state that "a large amount of VSVG was still localized on the Golgi even at 60 min after the temperature shift", and for image 12 they state "much more VSVG-GFP was observed on the Golgi at 60 min in Atg5/Ripk3DKO MEFs". Where is the difference? Why is the Golgi staining pattern so different in image 6? Where is the quantification of figure 8B?

3) Along these lines, I strongly recommend to clarify the quantifications of the immunofluorescence images. In most figure legends it is stated that "the population of cells with XY puncta was calculated". How was it calculated? Which software was used for counting the different puncta, and which algorithms were applied? The authors generally state that $n \geq 100$ cells were analyzed. In most images, only one single cell is shown. Why? The authors mainly use LAMP2 or mRFP-GFP detection in order to demonstrate ATG5-independent autophagy. With regard to LAMP2, they should clearly state what "large LAMP2 puncta" are. Which puncta size was used for this definition, and how was this quantified? Finally, co-localization of p-ULK1 and the employed Golgi markers should be quantified.

Minor points:

1) Is the immunopurification of ULK1 S746 sensitive to phosphatase treatment?

2) Figure 2A (untreated cells) and 2E: the LAMP2 staining is distributed across the entire image; in contrast, in figure 2B/C it is rather restricted to the shape/silhouette of the cell. How can this be explained?

In suppl. figure 5, the authors try to differentiate between trans, medial and cis-Golgi. I cannot see a clear difference between A/B and C. Again, the co-localization should be quantified.

Reviewer #3 (Remarks to the Author):

Using proteomics analysis, Torii and colleagues identified a novel phosphorylation site, S746, on Ulk1 that mediates alternative autophagy in response to the genotoxic agent etoposide. Using the S746A mutant to reconstitute Ulk1/2-deficient cells, the authors showed that this phosphorylation is specifically required for alternative autophagy, but not classical Atg5-dependent autophagy. P53 and the phosphatase PPMID dephosphorylates S637, an inhibitory site on Ulk1. The authors showed that this de-phosphorylation was required for S746 phosphorylation. Because the SSP motif within S746 is found in other RIP3 substrates, the authors speculated and confirmed that RIP3 is required for etoposide-induced Ulk1 phosphorylation at S746. RIPK3 phosphorylated Ulk1 at S746 in vitro, which was inhibited in the presence of RIPK3 kinase inhibitor. Accordingly, Ripk3 deficiency compromised etoposide-induced Ulk1 S746 phosphorylation and alternative autophagy. Reconstitution of Ripk3^{-/-} cells with wild type, but not kinase inactive RIPK3, restored alternative autophagy. However, RIPK3 is not required for Atg5/LC3-mediated classical autophagy. Mechanistically, the authors showed that while RIPK1, RIPK3 RHIM motif, RIPK3 phosphorylation at T231/S232, and the RIPK3 substrate MLKL were essential for necroptosis induced by TNF, cycloheximide and zVAD (TCZ), these necroptosis-associated events and adaptors are not required for RIPK3/Ulk1-mediated alternative autophagy. Finally, the authors showed that RIPK3-dependent alternative autophagy might have a role in degradation of Golgi-associated protein cargoes during genotoxic stress.

Overall, the results presented in the manuscript support a role of S746 phosphorylation in Ulk1-mediated alternative autophagy. The results also support a role for RIPK3 in mediating this Ulk1 phosphorylation event. Although the precise mechanism by which RIPK3 is activated in response to etoposide is unknown, the results do reveal a novel function of RIPK3 in regulatory alternative autophagy. These results should be of interest to the research community.

Specific comments:

1. A single genotoxic agent (etoposide) was used throughout the study. Do other genotoxic agents similarly stimulate ULK1 p-S746 in a RIPK3-dependent manner?
2. The constitutive interaction between RIPK3 and ULK1 is interesting (Fig. 3A). Do the authors have any insight which domain of RIPK3 is required for this interaction?
3. The system in Fig. 8 uses an artificial substrate VSVG to monitor Golgi to plasma membrane trafficking. It will be much more informative if the authors can demonstrate whether endogenous protein cargoes are indeed affected by alternative autophagy.
4. The protein RIPK3 should be capitalized throughout the text and legend.

Point-by-point Responses

Responses to the comments by Reviewer #1

Comment: In the manuscript by Torii et al., the authors identify a RIPK3-dependent phosphorylation site on ULK1 that is essential for ULK's ability to induce "alternate" autophagy at the Golgi. To do this, the authors develop a phospho-specific antibody and use KO lines and rescues with WT and phosphomutant ULK1, with alternate autophagy being measured by the presence of large LAMP structures or the lysosomal delivery and selective quenching of RFP-GFP. They also find that a prior dephosphorylation event, previously discovered by the group, is a prerequisite for RIPK3-ULK1 phosphorylation and that the mechanism of RIPK3 activation is distinct from the known necroptosis pathway. Finally, the authors present evidence that the alternate autophagy pathway is important for clearance of accumulated proteins that arise due to a block in Golgi export. On the whole the experiments are well executed and the data clear. I feel that this is an important piece of work in a pathway where very little is

known mechanistically. However, as the paper stands, many questions still remain (e.g. how is RIPK3 activated/ULK recruited to Golgi/consequences of this pathway etc.) and so this work represents more of an incremental step in answering these questions rather than a significant leap.

Response:

We greatly appreciate these constructive comments. In accordance with the comments, we extensively performed various experiments regarding (1) the mechanism of RIPK3 activation, (2) the mechanism of Ulk1 recruitment to the Golgi, and (3) the consequences of this pathway. We believe we have answered all of your comments fully.

To clarify the mechanism of RIPK3 activation, we focused on the upregulation of RIPK3 in response to etoposide treatment (Fig. 6I, lanes 2 and 3), and analyzed the involvement of p53-dependent transcriptional upregulation. By comparing *Atg5^{KO}* cells and *Atg5/p53^{DKO}* cells, we found substantially lower expression levels of *Ripk3* mRNA and RIPK3 protein in *Atg5/p53^{DKO}* cells upon etoposide treatment, indicating the crucial role of transcriptional regulation by p53 in etoposide-induced RIPK3 activation. These data were added to Fig. 8A and B, and described in the revised manuscript (page 15 line 5 - line 17). We added this pathway in our schematic model (Fig. 8G). We also performed mass-spectrometry analysis to detect post-translational modifications of RIPK3 in response to etoposide treatment, but, we found no RIPK3 modifications.

To clarify the mechanism of Ulk1 recruitment to the Golgi, we aimed to clarify the location at which Ulk1 is phosphorylated by RIPK3. To this end, we performed the close proximity (Duolink) assay between Ulk1 and RIPK3 and analyzed the location of interacting signals in etoposide-treated *Atg5^{KO}* cells. Interestingly, we found all the signals to be within the cytosol but not in the Golgi apparatus, indicating that Ulk1 phosphorylation occurs in the cytosol. We also performed the Duolink assay between Ulk1 and GS28, a golgi marker, in etoposide-treated *Atg5/Ulk1^{DKO}* cells. The results indicated that HA-Ulk1 (WT), but not the HA-Ulk1 (S746A) mutant, showed increased Duolink signals in etoposide-treated *Atg5/Ulk1^{DKO}* cells. Both of these results indicated that Ulk1 is phosphorylated in the cytosol and that this phosphorylation is essential for the recruitment of Ulk1 to the Golgi. We added these data to Fig. 8C-F and together with a description in the revised manuscript (page 15 line 18 - page 16 line 6).

To investigate the consequences of Ulk1 phosphorylation, we analyzed the molecules interacting with Ulk1. Immunoprecipitation experiments showed that the interaction between Ulk1 and FIP200 was decreased in etoposide-induced *Atg5^{KO}* cells, but not in *Atg5/RIPK3^{DKO}* cells. Consistent results were obtained when Atg13 was analyzed instead of Fip200. These data indicated that Ulk1 phosphorylation at Ser⁷⁴⁶ facilitates the dissociation of Ulk1 from Fip200 and Atg13. This is why this phosphorylation only contributes to alternative autophagy. We added these data to Fig. 8H, together with a description (page 16 line 7-14).

Comment #1: I am a little confused about the reasons for this alternate pathway and the consequences to a cell upon its loss. The authors uncover a very interesting cargo, related to Golgi trafficking, but what happens to the cell if this is blocked? Presumably, etoposide will eventually cause the cells to undergo apoptosis? but does loss of this alternate autophagy speed-up or delay this process? In their previous work (Torii et al., 2016 EMBOR) they proposed that loss of this (or a similar) pathway accelerated apoptosis due to reduced Noxa turnover? does loss of ULK phosphorylation have the same effect? Is the “Golgi trafficking defect” autophagy related to Noxa turnover or something different?

Response:

We believe this is an important point, and hence we analyzed the role of alternative autophagy on genotoxic stress-induced apoptosis. Comparison of wild-type thymocytes and *RIPK3^{KO}* thymocytes, in which alternative autophagy is blocked, demonstrated a similar extent of apoptosis upon various types of genotoxic stress (etoposide and camptothecin treatment). Noxa level was also not affected, and similar results were obtained when splenocytes were used. These data indicated that alternate autophagy does not affect apoptosis, at least in these cells. We added these data to Suppl. Fig. 22, and also added a description to the text (page 16, line 15 – page 17, line 2).

Genotoxic stress induces the cell death machinery. However, the extent of cell death depends on the extent of DNA damage. In our treatment of cells with genotoxic stress, most cells were alive at the time of the final analysis. In general, autophagy contributes to cellular remodeling in living cells upon cellular stress, and hence we analyzed the effects of alternative autophagy in living cells. Because alternative autophagy plays a role in Golgi trafficking, we tried to analyze the dynamics of VSVG-GFP and integrin $\alpha 5$ as an example of Golgi cargos. First, we demonstrated the induction of trafficking dysfunction upon etoposide treatment. Then, we observed the expression and

localization of VSVG and integrin $\alpha 5$. In *Atg5^{KO}* MEFs, expression levels of these molecules were not high after etoposide treatment, but were dramatically increased by the addition of E64d/pepstatin. Importantly, VSVG and integrin $\alpha 5$ were encircled by Lamp2 (or Lamp1) signals, indicating that undelivered VSVG-GFP and integrin $\alpha 5$ were engulfed into autolysosomes. In contrast, in *Atg5/RIPK3^{DKO}* MEFs, expression levels of VSVG-GFP and integrin $\alpha 5$ became high upon etoposide treatment, and were not increased by the addition of E64d/pepstatin. Importantly, these molecules were not merged with Lamp2 (or Lamp1) signals, indicating that VSVG-GFP and integrin $\alpha 5$ were not engulfed into autolysosomes. Because *Atg5/RIPK3^{DKO}* MEFs can perform canonical autophagy, but not alternative autophagy, undelivered Golgi cargos are degraded by alternative autophagy. This is hence an important cellular function of alternative autophagy. These data were added to Fig. 9, 10, and Suppl. Figs. 23-25, and also added a description to the text (page 17, line 2 – page 19, line 11).

Comment #2: Related to the above point, the authors have developed an elegant way to disrupt this pathway through ULK KO and then rescue with WT or phospho-mutant ULK1. Why did they not use this approach in the VSVG trafficking experiments? The authors instead use RIPK3 KO, and while this blocks alternate autophagy, it will/could also have other consequences (which could in turn be important for the trafficking).

Response:

We agree with this comment. We performed VSVG trafficking experiments using *Atg5/Ulk1^{DKO}* cell derivatives, and found that HA-Ulk1 (WT)-expressing *Atg5/Ulk1^{DKO}* MEFs and HA-Ulk1 (S746A)-expressing *Atg5/Ulk1^{DKO}* MEFs showed similar results to *Atg5^{KO}* MEFs and *Atg5/RIPK3^{DKO}* MEFs, respectively. These data confirmed that alternative autophagy contributes to the elimination of undelivered Golgi cargos upon genotoxic stress. These data were added to Suppl. Figs. 24 and 25, and described in the text (page 18, lines 13 – 18, page 19, lines 7 – 9).

Comment #3: Also related, it was not clear how the authors decided to look at Golgi trafficking. On page 14 the authors state that DNA damage generates excess proteins by disruption of Golgi to PM trafficking, and they postulate that autophagy may be clearing this build-up. But, where does these data come from that DNA damage blocks Golgi trafficking, as there are no references? Is it related to previously published GOLPH3 studies? If so, this is an obvious candidate for

the authors to analyse. Additionally, if it is DNA damage per se that is causing this pathway, then do the authors see the same phenomenon if another DNA-damaging agent is used (I do note that all the experiments here are done with etoposide)?

Response:

We previously demonstrated that alternative autophagy (also called Golgi-mediated degradation) is crucial for the degradation of undelivered cargos from the Golgi to the plasma membrane (EMBO J, 2016). Because genotoxic stress induces alternative autophagy, we suspected that Golgi trafficking may be inhibited by etoposide, which was clearly demonstrated in this study. The previously published GOLPH3 studies showed the induction of Golgi dispersion by genotoxic stress (Farber-Katz et al, 2014), and we described their data (page 17 lines 3 – 8) and cited this paper in the revised manuscript.

Furthermore, in accordance with the reviewer's suggestion, we analyzed the effect of different types of genotoxic stress, i.e., camptothecin and UV exposure, on the RIPK3-Ulk1 pathway. As expected, we obtained consistent results with etoposide treatment. These data were added to Suppl. Figs. 9, 22, and 23, and described in the text (page 10, lines 15 – 18, page 16, lines 20 – 21, page 18, lines 12 – 13).

Comment #4: From the methods, a little more detail on how the quantitation of alternate autophagy was carried out would be helpful. One method uses cells expressing mRFP-GFP and data is displayed as cells displaying red puncta. Does this mean cells with one puncta are counted the same as cells with 10 or more puncta? In other figures, alternate autophagy is assessed by number of cells displaying large LAMP2.

Response:

In accordance with this suggestion, we described detailed quantitation methods in the Method section. In the experiments using mRFP-GFP, autolysosomes are visualized as red puncta, because GFP fluorescence, but not RFP fluorescence, is decreased in acidic compartments. Therefore, the cellular RFP/GFP fluorescence ratio is increased in autophagy-induced cells. We acquired the images of RFP and GFP using Zen software, and analyzed cellular fluorescence intensity (intensity \times area/per cell) using Image J software. Then, we obtained the cellular RFP/GFP level of each cell.

Regarding the Lamp2 assay, we previously calculated the size of autophagic vacuoles by correlative light electron microscopy and found that large Lamp2 puncta (\geq

2 µm) were identical to autophagic vacuoles (Nature, 2009), and that there were few such big puncta in nonautophagic cells. Therefore, we regarded cells with more than one large LAMP2 puncta as autophagic cell. We described the detailed methods in the revised manuscript (page 26 line 15 - page 27 line 7).

Responses to comments by Reviewer #2

Comment: In the manuscript by Torii et al., the authors report that RIPK3-dependent phosphorylation of ULK1 at S746 is required for genotoxic stress-induced alternative autophagy. The group previously identified ATG5-independent autophagy. Here, they report that etoposide treatment results in ULK1 phosphorylation at S746. ULK1 phosphorylated at this site apparently localizes at the Golgi. The authors also report that S746 phosphorylation is only required for ATG5-independent autophagy, but not for canonical autophagy. Furthermore, the authors identify the necroptosis-mediating kinase RIPK3 as the kinase responsible for this phosphorylation. Finally, the authors report that p53/PPM1D-dependent dephosphorylation of ULK1 S637 is a prerequisite for RIPK3-dependent ULK1 phosphorylation. Although I think the authors made an interesting observation, I definitely think that this manuscript needs substantial revision in order to be acceptable for publication in NATURE COMMUNICATIONS. Especially, central mechanistic details remain unclear.

Response:

We greatly appreciate this high evaluation of our manuscript. Regarding the central mechanistic details, we performed several crucial experiments and added the following points to the manuscript: (1) the mechanism of RIPK3 activation upon genotoxic stress, (2) the mechanism of phospo-Ulk1 localization on the Golgi, (3) the role of alternative autophagy, as described below.

Major points:

Comment #1: The authors largely focus on the RIPK3-dependent phosphorylation of ULK1 at S746. Although this represents an interesting observation, mechanistic details upstream and downstream of this phosphorylation event remain entirely unclear. The authors report the etoposide-induced activation of RIPK3 does not resemble RIPK3 activation during necroptosis, i.e. it is independent of RIPK1, oligomerization or phosphorylation of RIPK3. But then, how is RIPK3 activated? Are there any kinases involved that become activated upon etoposide-induced

DNA-damage? With regard to ULK1, how does phosphorylation of S746 initiate ULK1 translocation to the Golgi? Or is this phosphorylation occurring at the Golgi? In suppl. Figure 4, it appears that the S746A-mutant can be readily transferred to the Golgi. Finally, does the phosphorylation of ULK1 at S746 alter its kinase activity or the interaction with known binding partners?

Response:

We greatly appreciate these constructive comments. Similar comments were also raised by Reviewer #1. In accordance with the comments, we extensively performed various experiments, particularly the clarification of (1) the upstream signal of RIPK3, (2) the association between ULK1 phosphorylation and Golgi recruitment, and (3) the effect of Ulk1 phosphorylation. We believe all these points are important and have been addressed in full.

To clarify the mechanism of RIPK3 activation, we focused on the upregulation of RIPK3 in response to etoposide treatment (Fig. 6I, lanes 2 and 3), and analyzed the involvement of p53-dependent transcriptional upregulation. By comparing *Atg5^{KO}* cells and *Atg5/p53^{DKO}* cells, we found much lower expression levels of *Ripk3* mRNA and RIPK3 protein in *Atg5/p53^{DKO}* cells upon etoposide treatment, indicating the crucial role of transcriptional regulation by p53 in etoposide-treated RIPK3 activation. These data were added to Fig. 8A and B, and described in the revised manuscript (page 15 line 5 - line 17). We added this pathway to our schematic model (Fig. 8G). We also performed mass-spectrometry analysis to detect post-translational modifications of RIPK3 in response to etoposide treatment, but found no RIPK3 modifications.

To clarify the mechanism of Ulk1 recruitment to the Golgi, we aimed to clarify the location at which Ulk1 is phosphorylated by RIPK3. To this end, we performed the close proximity (Duolink) assay between Ulk1 and RIPK3 and analyzed the location of the interacting signals in etoposide-treated *Atg5^{KO}* cells. Interestingly, we found all the signals within the cytosol but not in the Golgi apparatus, indicating that Ulk1 phosphorylation occurs in the cytosol. We also performed the Duolink assay between Ulk1 and GS28, a Golgi marker, in etoposide-treated *Atg5/Ulk1^{DKO}* cells. These results demonstrated that HA-Ulk1 (WT), but not the HA-ULK1 (S746A) mutant, showed increased Duolink signals in etoposide-treated *Atg5/Ulk1^{DKO}* cells. Both of these results indicate that Ulk1 is phosphorylated in the cytosol and that this phosphorylation is essential for the recruitment of Ulk1 to the Golgi. We added these data to Fig. 8C-F and described it in the revised manuscript (page 15 line 18 - page 16 line 6).

To investigate the effects of the phosphorylation of ULK1 at S746, we analyzed molecules known to interact with Ulk1. The immunoprecipitation assay showed that interaction between Ulk1 and FIP200 was decreased in etoposide-induced *Atg5^{KO}* cells, but not in *Atg5/RIPK3^{DKO}* cells. Consistent results were obtained when Atg13 was analyzed instead of Fip200. These data indicated that the phosphorylation of Ulk1 at Ser⁷⁴⁶ facilitates the dissociation of Ulk1 from Fip200 and Atg13. This is why this phosphorylation only contributes to alternative autophagy. We added these data to Fig. 8H, and added a description to the text (page 16 line 7–14). Regarding the effects of the phosphorylation of Ser⁷⁴⁶ of Ulk1 on its kinase activity, phosphorylation may not affect kinase activity, because this residue is located in an intrinsically disordered region.

Comment #2: In figure 8, the authors try to establish the biological relevance of ATG5-independent autophagy. They suggest that ULK1 phosphorylation at S746 is important for the elimination of superfluous undelivered proteins from the Golgi. Although I agree that VSVG-GFP might co-localize with lysosomes and that VSVG-GFP expression is increased by E64d/Pepstatin, I think it is difficult to draw any conclusions from the VSVG trafficking assay. For example, for image 6 (Figure 8B) the authors state that “a large amount of VSVG was still localized on the Golgi even at 60 min after the temperature shift”, and for image 12 they state “much more VSVG-GFP was observed on the Golgi at 60 min in *Atg5/RIPK3DKO* MEFs”. Where is the difference? Why is the Golgi staining pattern so different in image 6? Where is the quantification of figure 8B?

Response:

We greatly appreciate this comment. We apologize for confusing the reviewer owing to our unclear images and description. We replaced some of the images to more representative ones. In image 6, VSVG-GFP was located on the Golgi, as described in the original manuscript. In image 12, VSVG-GFP was localized on the Golgi and also in the cytosol.

Therefore, we included the descriptions "etoposide treatment suppressed this delivery, because of the low expression of PM-localized VSVG-GFP even at 60 min after the temperature shift (Fig. 9B, image 6). In contrast to *Atg5^{KO}* MEFs, in *Atg5/RIPK3^{DKO}* MEFs, VSVG-GFP was observed not only on the Golgi but also in the cytoplasmic punctate structures at 60 min (Fig. 9B, compare images 6 and 12). " (page 17 line 15– page 18 line 5)

Comment #3: Along these lines, I strongly recommend to clarify the quantifications of the immunofluorescence images. In most figure legends it is stated that “the population of cells with XY puncta was calculated”. How was it calculated? Which software was used for counting the different puncta, and which algorithms were applied? The authors generally state that $n \geq 100$ cells were analyzed. In most images, only one single cell is shown. Why? The authors mainly use LAMP2 or mRFP-GFP detection in order to demonstrate ATG5-independent autophagy. With regard to LAMP2, they should clearly state what “large LAMP2 puncta” are. Which puncta size was used for this definition, and how was this quantified? Finally, co-localization of p-ULK1 and the employed Golgi markers should be quantified.

Response:

Thank you very much for the helpful comments. Similar comments were also raised by Reviewer #1. In accordance with this suggestion, we reconsidered the quantitation methods and described our detailed quantitation methods in the Methods section. In the experiments using mRFP-GFP, autolysosomes were visualized as red puncta, because GFP fluorescence, but not RFP fluorescence, is decreased in acidic compartments. Therefore, the cellular RFP/GFP fluorescence ratio is increased in autophagy-induced cells. We acquired RFP and GFP images using Zen software, and analyzed cellular fluorescence intensity (intensity \times area/per cell) using Image J software. Then, we obtained cellular RFP/GFP levels of each cell.

Regarding the Lamp2 assay, we previously calculated the size of autophagic vacuoles by correlative light electron microscopy and found that large Lamp2 puncta ($\geq 2 \mu\text{m}$) were identical to autophagic vacuoles (Nature, 2009), and that there were very few big puncta in nonautophagic cells. Therefore, we regarded cells with more than one large LAMP2 ($\geq 2 \mu\text{m}$) puncta as autophagic cells, and analyzed cells using Zen software (Zeiss). We described the detailed methods of these experiments in the revised manuscript (page 26 line 15 - page 27 line 7).

We showed only one cell in our data of various experiments because of space limitation and data capacity. To provide more data, we also showed quantification data for all the images. In the revised manuscript, we further added a representative image with multiple cells as Suppl. Fig. 8B.

Regarding the colocalization of p-Ulk1 and Golgi markers, we analyzed this using the Duolink assay and found that p-Ulk1⁷⁴⁶ preferentially translocates to the Golgi.

We added these data to Figs. 8E and 8F, and described it in the revised manuscript (page 16 lines 1 - 6).

Minor points:

Comment #4: Is the immunopurification of ULK1 S746 sensitive to phosphatase treatment?

Response:

In accordance with this suggestion, we performed the phosphatase treatment assay. As described, phosphatase treatment decreased the immunopurification of phosphorylated Ulk1. These data were added to Suppl. Fig. 2, and described it in the revised manuscript (page 7 lines 6 - 9).

Comment #5: Figure 2A (untreated cells) and 2E: the LAMP2 staining is distributed across the entire image; in contrast, in figure 2B/C it is rather restricted to the shape/silhouette of the cell. How can this be explained? In suppl. figure 5, the authors try to differentiate between trans, medial and cis-Golgi. I cannot see a clear difference between A/B and C. Again, the co-localization should be quantified.

Response:

We greatly appreciate these helpful comments. In the original manuscript, we did not show the cell shape, and hence it might be difficult to distinguish one cell from the neighboring cells. Therefore, in the revised manuscript, we have indicated the cell shape using a dashed line. Another reason is cellular variety. To avoid confusion, we removed the Lamp2 images of Fig. 2C and 2D from the original manuscript, as these images are not required for our conclusion.

According to the Reviewer's comment regarding quantification of ULK1 localization on the Golgi, we performed the close proximity assay between Ulk1 and GS28, a pan-Golgi marker. Quantitation of the results demonstrated the importance of Ulk1 phosphorylation at Ser⁷⁴⁶ for Golgi translocation. We added these data to Fig. 8E and F, and described it in the revised manuscript (page 16 lines 1 - 6).

Responses to comment by Reviewer #3

Comment : Using proteomics analysis, Torii and colleagues identified a novel phosphorylation site, S746, on Ulk1 that mediates alternative autophagy in response to the genotoxic agent etoposide. Using the S746A mutant to reconstitute Ulk1/2-deficient cells, the authors showed that this phosphorylation is specifically required for alternative autophagy, but not classical Atg5-dependent autophagy. P53 and the phosphatase PPMID dephosphorylates S637, an inhibitory site on Ulk1. The authors showed that this de-phosphorylation was required for S746 phosphorylation. Because the SSP motif within S746 is found in other RIP3 substrates, the authors speculated and confirmed that RIP3 is required for etoposide-induced Ulk1 phosphorylation at S746. RIPK3 phosphorylated Ulk1 at S746 in vitro, which was inhibited in the presence of RIPK3 kinase inhibitor. Accordingly, *Ripk3* deficiency compromised etoposide-induced Ulk1 S746 phosphorylation and alternative autophagy. Reconstitution of *Ripk3*^{-/-} cells with wild type, but not kinase inactive RIPK3, restored alternative autophagy. However, RIPK3 is not required for Atg5/LC3-mediated classical autophagy. Mechanistically, the authors showed that while RIPK1, RIPK3 RHIM motif, RIPK3 phosphorylation at T231/S232, and the RIPK3 substrate MLKL were essential for necroptosis induced by TNF, cycloheximide and zVAD (TCZ), these necroptosis-associated events and adaptors are not required for RIPK3/Ulk1-mediated alternative autophagy. Finally, the authors showed that RIPK3-dependent alternative autophagy might have a role in degradation of Golgi-associated protein cargoes during genotoxic stress. Overall, the results presented in the manuscript support a role of S746 phosphorylation in Ulk1-mediated alternative autophagy. The results also support a role for RIPK3 in mediating this Ulk1 phosphorylation event. Although the precise mechanism by which RIPK3 is activated in response to etoposide is unknown, the results do reveal a novel function of RIPK3 in regulatory alternative autophagy. These results should be of interest to the research community.

Response:

We greatly appreciate the high evaluation of our manuscript. In accordance with the reviewer's comment regarding the upstream mechanism of RIPK3 activation upon genotoxic stress, we focused on RIPK3 upregulation in response to etoposide treatment (Fig. 6J, lanes 2 and 3), and analyzed the involvement of p53-dependent transcriptional upregulation. By comparing *Atg5*^{KO} cells and *Atg5/p53*^{DKO} cells, we found much lower expression levels of *Ripk3* mRNA and RIPK3 protein in *Atg5/p53*^{DKO} cells upon

etoposide treatment, indicating the crucial role of transcriptional regulation by p53 in etoposide-treated RIPK3 activation. These data were added to Fig. 8A and B, and described in the revised manuscript (page 15 line 5 - line 17). We added this pathway to the schematic model (Fig. 8G). We also performed mass-spectrometry analysis to detect post-translational modifications of RIPK3 in response to etoposide treatment, but found no RIPK3 modifications.

Comment #1: A single genotoxic agent (etoposide) was used throughout the study. Do other genotoxic agents similarly stimulate ULK1 p-S746 in a RIPK3-dependent manner?

Response:

We believe this is an important point. In accordance with the reviewer's suggestion, we analyzed the effects of different types of genotoxic stress, i.e., camptothecin and UV exposure, on the RIPK3-Ulk1 pathway. As expected, we obtained consistent results with those of etoposide treatment. These data were added to Suppl. Figs. 9, 22, and 23, and described in the text (page 10, lines 15 – 18, page 16, lines 20 – 21, page 18, lines 12 – 13).

Comment #2: The constitutive interaction between RIPK3 and ULK1 is interesting (Fig. 3A). Do the authors have any insight which domain of RIPK3 is required for this interaction?

Response:

We appreciate this constructive comment. As suggested, we generated several RIPK3 deletion mutants and analyzed their interaction with Ulk1. As the N-terminal deletion mutant lacking the kinase domain did not interact with Ulk1, the kinase domain is thought to be important for this interaction. These data were added to Suppl. Fig. 16, and described in the text (page 13, lines 19 – 22).

Comment #3: The system in Fig. 8 uses an artificial substrate VSVG to monitor Golgi to plasma membrane trafficking. It will be much more informative if the authors can demonstrate whether endogenous protein cargoes are indeed affected by alternative autophagy.

Response:

We appreciate this very important comment. In accordance with this suggestion, we analyzed endogenous integrin alpha5 as a Golgi cargo and obtained following findings: In *Atg5^{KO}* MEFs, Golgi trafficking was disrupted by etoposide, but undelivered integrin alpha5 was degraded by alternative autophagy. In contrast, in *Atg5/RIPK3^{DKO}* MEFs, undelivered integrin alpha5 was not degraded and accumulated in the cytosol. This was confirmed because cytoplasmic integrin alpha5 appeared upon the inhibition of autophagic degradation by E64d/pepstatin in etoposide-treated *Atg5^{KO}* MEFs. Consistently, etoposide-induced cytoplasmic integrin alpha5 puncta were observed in *Atg5/Ulk1^{DKO}* MEFs expressing the HA-Ulk1 (S746A) mutant, but not in those expressing HA-Ulk1 (WT). These data indicate that alternative autophagy contributes to the elimination of undelivered endogenous Golgi cargoes upon genotoxic stress. These data were added to Fig. 10 and Suppl. Fig. 25, and described in the text (page 18, line 19 –page 19, line 9).

Comment #4: The protein RIPK3 should be capitalized throughout the text and legend.

We appreciate this kind comment. RIPK3 was capitalized throughout the manuscript.

Reviewers' comments:

Reviewer #1 (Remarks to the Author):

The authors have done a very good job in addressing my comments and I have no further concerns.

Reviewer #2 (Remarks to the Author):

In the revised version of the manuscript by Torii et al., the authors identified a novel phosphorylation site of ULK1 that is required for genotoxic stress-induced alternative autophagy. Once more I would like to point out that I am convinced that the authors make an important observation, and this work complements the understanding of ULK1 signaling. I also appreciate that the authors attempted to address my previous comments. There are only some minor points that remain. If these issues can be addressed, the manuscript should be acceptable for publication in NATURE COMMUNICATIONS. In my comments below, I am referring to my previous numbering.

Major points:

- 1) The authors show that RIPK3 activation upon genotoxic stress differs from canonical RIPK3 activation during necroptosis, i.e. it is independent of RIPK1 or apparent post-translational modifications of RIPK3. The authors mainly focus on p53-dependent transcriptional up-regulation of RIPK3. It would be interesting to know if the sole inducible up-regulation of RIPK3 is sufficient to induce ULK1 phosphorylation at S746 for example in cells expressing the ULK1 variant S637A. This would further strengthen the authors' conclusion that p53-dependent regulation of RIPK3 expression is sufficient to induce alternative autophagy. With regard to the "canonical" interacting components of ULK1, the authors have performed co-immunoprecipitation studies (new figure 8H). I think this aspect needs some additional experiments in order to make a valid conclusion. For example, the authors could analyze the interaction of the S746E/A variants with ATG13 and/or FIP200. Additionally, they could perform immunofluorescence and/or proximity ligation assays in order to convincingly show that ATG13/FIP200 are not recruited to the Golgi.
- 2) Ok
- 3) Ok

Minor points:

- 4) Ok
- 5) Ok

Reviewer #3 (Remarks to the Author):

In the revised manuscript, the authors did an outstanding job addressing all the concerns raised in the original review. Specifically, the authors have now provided data supporting (1) a role for p53 in inducing RIPK3 expression and activation in response to etoposide, (2) that other genotoxic stress such as UV and camptothecin also induced ULK1 phosphorylation at S746 and alternative autophagy in a RIPK3-dependent manner, (3) that this alternative autophagy pathway is responsible for clearance of an endogenous cargo integrin alpha 5. As a result, the revised manuscript presents a clear and strong case that the RIPK3-ULK1 axis regulates alternative autophagy.

Point-by-point Responses to the Reviewers comments

Responses to the comments by Reviewer #1

The authors have done a very good job in addressing my comments and I have no further concerns.

Response:

We greatly appreciate this comment.

Responses to the comments by Reviewer #2

Comment #1: In the revised version of the manuscript by Torii et al., the authors identified a novel phosphorylation site of ULK1 that is required for genotoxic stress-induced alternative autophagy. Once more I would like to point out that I am convinced that the authors make an important observation, and this work complements the understanding of ULK1 signaling. I also appreciate that the authors attempted to address my previous comments. There are only some minor points that remain. If these issues can be addressed, the manuscript should be acceptable for publication in NATURE COMMUNICATIONS. In my comments below, I am referring to my previous numbering.

Response:

We greatly appreciate this high evaluation of our manuscript, as well as the helpful and important comments. In accordance with the comments, we performed several experiments and believe all the points have been addressed in full in the revised version of the manuscript.

Points#1: The authors show that RIPK3 activation upon genotoxic stress differs from canonical RIPK3 activation during necroptosis, i.e. it is independent of RIPK1 or apparent post-translational modifications of RIPK3. The authors mainly focus on p53-dependent transcriptional up-regulation of RIPK3. It would be interesting to know if the sole inducible up-regulation of RIPK3 is sufficient to induce ULK1 phosphorylation at S746 for example in cells expressing the ULK1 variant S637A. This would further strengthen the authors' conclusion that p53-dependent regulation of RIPK3 expression is sufficient to induce alternative autophagy.

Response:

We showed that Ulk1⁷⁴⁶ phosphorylation occurs via two steps, i.e., the first step is from phospho-Ulk1⁶³⁷ to dephospho-Ulk1⁶³⁷ via PPM1D, and the second step is from dephospho-Ulk1⁶³⁷ to dephospho⁶³⁷/phospho⁷⁴⁶-Ulk1 via Ripk3. The reviewer asked whether p53-dependent upregulation of Ripk3 is sufficient to activate the second step.

To answer this question, we expressed Ulk1 (S637A) in Atg5/Ulk1 DKO MEFs, and further transfected Ripk3, instead of subjecting the cells to genotoxic stress. After 24 hr, we observed marked phspho-Ulk1⁷⁴⁶ signals in the Golgi. Furthermore, this Ulk1⁷⁴⁶ phosphorylation was not observed when the kinase-dead mutant of Ripk3 (Ripk3 K51A) was expressed (Suppl. Fig. 21B and C). From these data, we concluded that the upregulation of Ripk3 is sufficient to activate the second step (Ulk1 phosphorylation at Ser⁷⁴⁶). We added these results to the revised manuscript, and added the following description (page 15, lines 15–18).

“The importance of this RIPK3 upregulation was confirmed by the results that the simple expression of RIPK3, but not that of kinase-deficient RIPK3, was sufficient in generating p-Ulk1⁷⁴⁶ in HA-Ulk1 (S637A)-expressing *Atg5/Ulk1*^{DKO} MEFs (Suppl. Fig. 21B, C).”

Points#2: With regard to the “canonical” interacting components of ULK1, the authors have performed co-immunoprecipitation studies (new figure 8H). I think this aspect needs some additional experiments in order to make a valid conclusion. For example, the authors could analyze the interaction of the S746E/A variants with ATG13 and/or FIP200.

Response:

In the original manuscript, we showed that ATG13 and FIP200 interact with Ulk1 in *Atg5/Ripk3*^{DKO} MEFs, but not in *Atg5*^{KO} MEFs (Fig. 8H). This difference is thought to be owing to the difference in Ulk1 Ser⁷⁴⁶ phosphorylation status, because Ulk1 is phosphorylated at Ser⁷⁴⁶ in *Atg5*^{KO} MEFs, but not in *Atg5/Ripk3*^{DKO} MEFs. The reviewer pointed out that this point should be confirmed using Ulk1 S746D/A mutants.

In accordance with this suggestion, we analyzed the interaction between Ulk1 (S746D) and Ulk1 (S746A) with ATG13/FIP200 during etoposide treatment. As expected, Ulk1 (S746A) interacted with ATG13/FIP200 in untreated and etoposide-treated MEFs, whereas this interaction was not observed when using Ulk1

(S746D). From these data, we concluded that the interaction of Ulk1 with ATG13/FIP200 is regulated by Ulk1 Ser⁷⁴⁶ phosphorylation status. We added these results to Suppl. Fig. 22A, and described them in the manuscript, as follows (page 16, lines 17–220).

“Consistently, when we analyzed the interaction of HA-Ulk1 (S746D) and HA-Ulk1 (S746A) with Atg13/Fip200 in *Atg5/Ulk1^{DKO}* MEFs upon etoposide treatment, we observed a positive interaction for HA-Ulk1 (S746A), but not for HA-Ulk1 (S746D) (Suppl. Fig. 22A).”

Points#3: Additionally, they could perform immunofluorescence and/or proximity ligation assays in order to convincingly show that ATG13/FIP200 are not recruited to the Golgi.

Response:

In the original manuscript, we showed that phospho-Ulk1⁷⁴⁶ localized mainly to the Golgi and did not interact with ATG13 and FIP200. Therefore, ATG13 and FIP200 are expected not to be recruited to the Golgi. The reviewer suggested that we should confirm this expectation.

Therefore, in accordance with the suggestion, we performed a proximity ligation assay between FIP200/ATG13 and the Golgi, and quantified their interaction. As expected, the interaction between Ulk1 and the Golgi was observed upon etoposide treatment (Fig. 8E, F), but the interaction of FIP200/ATG13 with the Golgi was not observed. We added these results to Suppl. Fig. 22B and C, and described them in the revised manuscript, as follows (page 16 lines 20–22).

“Furthermore, Fip200 and Atg13 did not localized to the Golgi in etoposide-treated *Atg5^{KO}* MEFs (Suppl. Fig. 22B and C).”

Responses to the comments by Reviewer #3

In the revised manuscript, the authors did an outstanding job addressing all the concerns raised in the original review. Specifically, the authors have now provided data supporting (1) a role for p53 in inducing RIPK3 expression and activation in response to etoposide, (2) that other genotoxic stress such as UV and camptothecin also induced ULK1 phosphorylation at S746 and alternative autophagy in a RIPK3-dependent manner, (3) that this alternative autophagy pathway is

responsible for clearance of an endogenous cargo integrin alpha 5. As a result, the revised manuscript presents a clear and strong case that the RIPK3-ULK1 axis regulates alternative autophagy.

Response:

We greatly appreciate this high evaluation of our manuscript.